# Adaptation to glucose starvation is associated with molecular reorganization of the circadian clock in *Neurospora crassa*

Anita Szőke[1], Orsolya Sárkány[1], Géza Schermann[2], Orsolya Kapuy[3], Axel CR Diernfellner[4], Michael Brunner[4], Norbert Gyöngyösi[3†], Krisztina Káldi[1*†]

[1]Department of Physiology, Semmelweis University, Budapest, Hungary; [2]Department of Neurovascular Cellbiology, University Hospital Bonn, Bonn, Germany; [3]Department of Molecular Biology, Semmelweis University, Budapest, Hungary; [4]Biochemistry Center, Heidelberg University, Heidelberg, Germany

**Abstract** The circadian clock governs rhythmic cellular functions by driving the expression of a substantial fraction of the genome and thereby significantly contributes to the adaptation to changing environmental conditions. Using the circadian model organism *Neurospora crassa*, we show that molecular timekeeping is robust even under severe limitation of carbon sources, however, stoichiometry, phosphorylation and subcellular distribution of the key clock components display drastic alterations. Protein kinase A, protein phosphatase 2 A and glycogen synthase kinase are involved in the molecular reorganization of the clock. RNA-seq analysis reveals that the transcriptomic response of metabolism to starvation is highly dependent on the positive clock component WC-1. Moreover, our molecular and phenotypic data indicate that a functional clock facilitates recovery from starvation. We suggest that the molecular clock is a flexible network that allows the organism to maintain rhythmic physiology and preserve fitness even under long-term nutritional stress.

*For correspondence:
kaldi.krisztina@med.semmelweis-univ.hu

†These authors contributed equally to this work

**Competing interest:** The authors declare that no competing interests exist.

## Editor's evaluation

This manuscript is of interest to researchers working in the areas of chronobiology, metabolism, or environmental adaptation mechanisms. The authors show that starvation decreases the abundance of the fungal circadian clock component white collar complex (WCC). However, neither phase nor the amplitude of the RNA oscillation of the critical circadian clock gene *frq* are affected by starvation, indicating a mechanism that recalibrates the central clockwork. Furthermore, *Neurospora* recovers faster from starvation in the presence of a functioning clock, adding further evidence for the importance of the circadian clock for organismal fitness.

## Introduction

Circadian clocks are endogenous timekeeping systems allowing organisms to adapt to cyclic changes in the environment. *Circadian clocks rely on* transcriptional-translational feedback loops (TTFL) in which positive elements of the machinery activate the expression of oscillator proteins which in turn negatively feed back on their own transcription. Circadian clocks are closely linked to metabolism. On the one hand, the clock rhythmically modulates many metabolic pathways (*Asher and Schibler, 2011*; *Huang et al., 2011*; *Bray and Young, 2011*; *Duez and Staels, 2009*), and on the other hand, nutrients

and metabolic cues influence clock function (e.g. *Roenneberg and Rehman, 1996*; *Johnson, 1992*; *Stokkan et al., 2001*). It is therefore not surprising that in human, conditions involving circadian rhythm dysfunction, such as shift work or jetlag, are associated with an increased risk of metabolic disorders including obesity, metabolic syndrome and type 2 diabetes (*Baron and Reid, 2014*).

Although the timing of nutrient availability as an important Zeitgeber determines the phase of the rhythm in many organisms, the circadian oscillator was found to run with constant speed and thus to maintain a constant period in the model organisms *Neurospora crassa* and *Synechococcus elongatus* (*Sancar et al., 2012*; *Johnson and Egli, 2014*). Accurate synchronization of metabolic processes with recurrent environmental conditions, such as light-darkness or temperature fluctuations, may be particularly critical for efficient adaptation to nutrient deprivation. However, because glucose levels were shown to affect many signal transduction pathways as well as transcription and translation rates in different organisms (*Ashe et al., 2000*; *Corral-Ramos et al., 2021*; *Jona et al., 2000*; *Sancar et al., 2012*), glucose deficiency may challenge the TTFL-based circadian clock to operate at a constant period. Molecular mechanisms of nutrient compensation have been intensively investigated in *Neurospora crassa*. In *Neurospora* the White-Collar-Complex (WCC) composed of the GATA-type transcription factors WC-1 and WC-2, and Frequency (FRQ) represent the core components of the circadian clock. The WCC supports expression of FRQ which then interacts with an RNA helicase (FRH) and the casein kinase 1a (CK1a) (*Cheng et al., 2005*; *Görl et al., 2001*). The FRQ-FRH-CK1a complex acting as the negative factor of the clock facilitates phosphorylation and thus inhibition of the WCC. During a circadian day FRQ is progressively phosphorylated, which reduces its inhibitory potential and leads to its degradation (*Querfurth et al., 2011*). The negative feedback and the gradual maturation of FRQ together result in rhythmic changes of WCC activity and *frq* levels (*Larrondo et al., 2015*; *Querfurth et al., 2011*; *Schafmeier et al., 2006*). The negative feedback loop is connected to a positive loop, in which FRQ supports the accumulation of both WC-1 and WC-2 (*Cheng et al., 2001*; *Lee et al., 2000*). Similarly to other organisms, the *Neurospora* circadian clock supports rhythmic expression of about 10% of the genome (*Hurley et al., 2014*; *Sancar et al., 2015*). The WCC as the major photoreceptor of *Neurospora* is activated by light and thereby transduces light information to the clock (*Froehlich et al., 2002*; *He et al., 2002*). In *Neurospora*, short-term (0–16 hr) glucose deprivation triggers compensatory mechanisms at the transcriptional and posttranscriptional levels that maintain expression levels of the core clock proteins, thereby keeping period length constant (*Adhvaryu et al., 2016*; *Emerson et al., 2015*; *Gyöngyösi et al., 2017*; *Olivares-Yañez et al., 2016*).

Aim of this study was to characterize how chronic glucose deprivation affects the molecular clock and what role the circadian clock plays in the adaptation to starvation. We analyzed the transcriptome response to long-term glucose starvation in *wt* and the clock-less mutant *Δwc-1* and found that the WCC has a striking impact on nutrient-dependent expression of a large set of genes, including enzymes and regulators of carbohydrate, amino acid, and fatty acid metabolism. We show that molecular timekeeping is robust even under severe limitation of carbon sources. Moreover, our data provide evidence that the TTFL is able to function in a wide range of stoichiometric conditions of its key elements, dependent on glucose availability. Our results show that *Neurospora* recovers faster from starvation in the presence of a functioning clock, suggesting a significant impact of the circadian clock on organismal fitness.

## Results

### Glucose-deprivation results in altered expression of core clock components

To assess how long-term glucose starvation might affect the expression of clock components, liquid cultures of *wt Neurospora* grown at 2% glucose (standard medium) were transferred to a starvation medium with 0.01% glucose for 40 hr. Cultures were kept in constant light (LL). In LL, both the negative and the positive feedback persist, however, the circadian oscillation stops and both RNA and protein levels of the core clock components are at steady state (*Crosthwaite et al., 1995*; *Elvin et al., 2005*). Therefore, phase effects that might arise in response to different manipulations under free-running conditions can be excluded. The growth of *Neurospora* virtually stopped after the medium change and the expression of the major clock components was characteristically changed compared to standard cultures (*Figure 1A and B*). Both WC-1 and WC-2 expression decreased gradually to

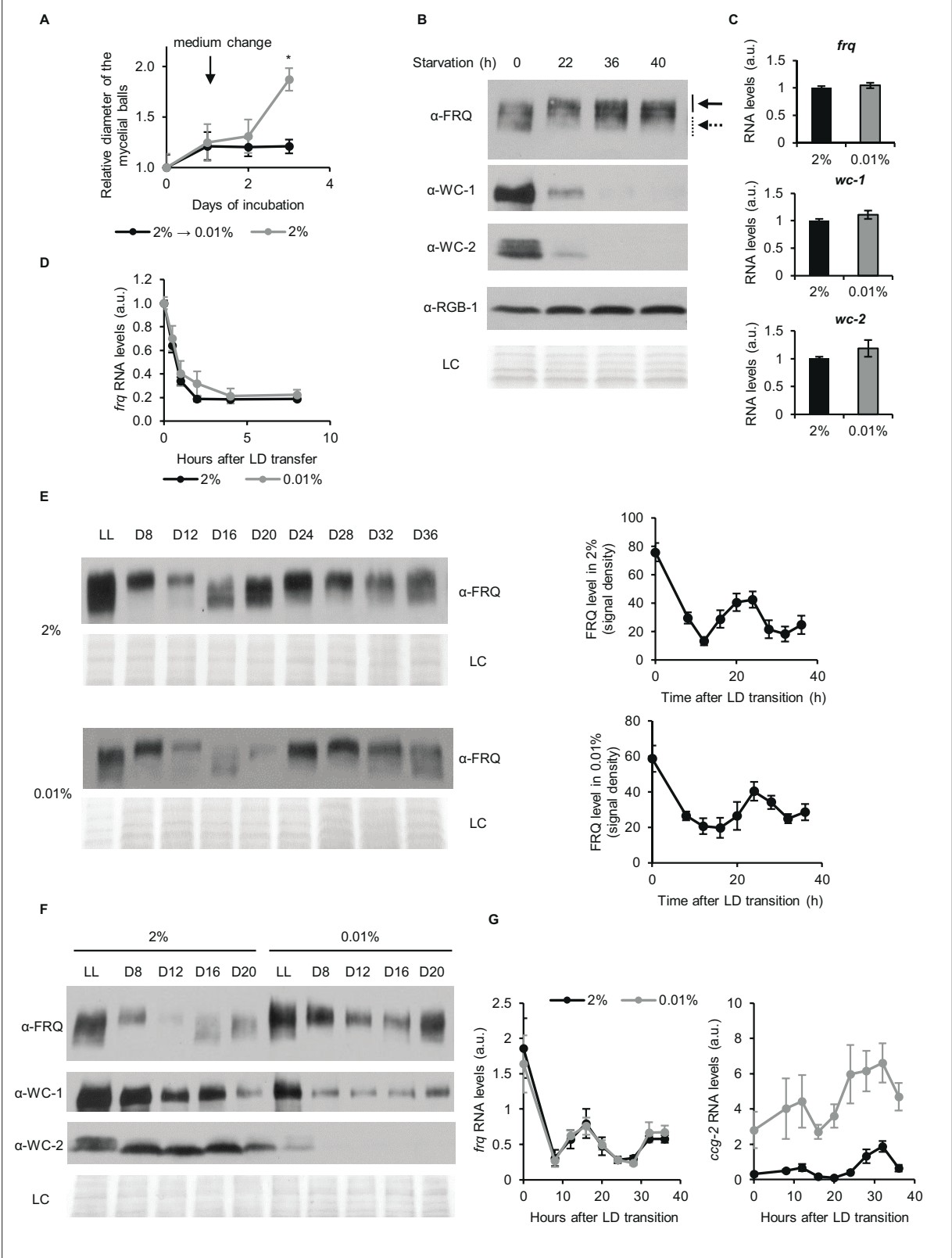

**Figure 1.** Despite changes of the stoichiometry of clock components circadian time-measuring is sustained upon glucose depletion. (**A**) *Neurospora* growth is arrested in starvation medium. Following an incubation for 24 hr in standard liquid medium, mycelia were transferred to media containing either 2% or 0.01% glucose. Diameter of the mycelial balls was measured each day. The arrow indicates the time of medium change. (n=3, ± SEM, Repeated measures ANOVA, significant time-treatment interaction; post hoc analysis: Fisher LSD test). (**B**) Long-term glucose starvation affects

*Figure 1 continued on next page*

*Figure 1 continued*

expression of clock proteins in *wt*. Mycelial discs were incubated for 24 hr in standard liquid medium and then transferred to starvation medium (time point 0). Samples were harvested at the indicated time points. Cell extracts were analyzed by western blotting. Solid and dashed arrows indicate hyper- and hypophosphorylated forms of FRQ, respectively. RGB-1 and Ponceau staining (LC: loading control) are shown as loading controls. (n=3) See also **A**. (**C**) RNA levels of *frq*, *wc-1* and *wc-2* are similar under standard and nutrient limited conditions. Mycelial discs of the *wt* strain were incubated in standard liquid medium for 24 hr, then transferred to fresh media containing either 0.01% or 2% glucose and incubated for 40 hr in LL. RNA levels were normalized to that in cells grown in standard medium. (n=9–22, ± SEM, two-sample t-test, n.s.). (**D**) Stability of *frq* RNA is not affected by starvation. Growth conditions were described in (**C**). Following 40 hr of incubation in LL, cultures were transferred to DD (time point 0). Samples were harvested at the indicated time points. RNA levels were normalized to those measured at time point 0. (n=6, ± SEM, Repeated measures ANOVA, n.s.). (**E**) Left panel: FRQ level oscillates under starvation conditions in DD. Following an incubation in standard liquid medium for 24 hr, mycelia were transferred to standard or starvation medium. After 24 hr incubation in LL, cultures were transferred to DD. Samples were harvested at the indicated time points. (n=3, LC: loading control) Right panel: FRQ specific signals were analyzed by densitometry. (n=4–6). (**F**) WC levels are reduced and FRQ is hyperphosphorylated during glucose starvation. The experiment was performed as described in (**E**). Cell extracts from both growth conditions were analyzed on the same gel. (n=3, LC: loading control). (**G**) Expression of *frq* and *ccg-2* is rhythmic during long-term glucose starvation. Experiment was performed as described in (**E**). RNA levels were determined by qPCR. (n=3–11, ± SEM, Repeated measures ANOVA, n.s.).

The online version of this article includes the following source data and figure supplement(s) for figure 1:

**Source data 1.** Source data for *Figure 1B*.

**Source data 2.** Source data for *Figure 1E*.

**Source data 3.** Source data for *Figure 1F*.

**Source data 4.** Actin levels are decreased in glucose starvation.

**Figure supplement 1.** Analysis of expression changes in clock components.

**Figure supplement 1—source data 1.** Source data for *Figure 1—figure supplement 1B*.

**Figure supplement 2.** The circadian output is rhythmic in starvation.

about 15% and 20% of the initial levels, respectively (*Figure 1—figure supplement 1A*). The amount of FRQ remained relatively constant after glucose deprivation, but a mobility shift characteristic for hyperphosphorylation of the protein was observed (*Figure 1B*).

Because hyperphosphorylated FRQ exerts a reduced negative feedback on WCC (*Schafmeier et al., 2006*), we hypothesized that the starvation-induced phosphorylation of FRQ might lead to an increase in WCC activity and consequently to the acceleration of its decay (*Kodadek et al., 2006*; *Punga et al., 2006*; *Schafmeier et al., 2008*). Hence, we followed WC levels in cultures treated with the translation inhibitor cycloheximide to assess WCC stability (*Figure 1—figure supplement 1B*). Our data suggest that increased turnover of WC-1 may be, at least partially, responsible for the low WCC levels in the starved cells.

In LL WCC constantly promotes transcription of *frq* and *wc-1*. Although WCC levels were significantly different under standard and glucose-starved conditions, RNA levels of *frq*, *wc-1 and wc-2* were similar (*Figure 1C*), suggesting a compensatory mechanism that either maintains the active pool of WCC under various nutritional conditions constant or stabilizes the RNA. In the next experiment we examined *frq* RNA levels after a light-dark transfer (LD), when transcription of *frq* is repressed, and therefore changes in *frq* levels reflect RNA degradation. *frq* RNA levels after LD transition were similar under both culture conditions, suggesting that changes in RNA stability do not contribute to the maintenance of *frq* levels during starvation (*Figure 1D*).

Although the expression of FRQ and WCC is interdependent (*Cheng et al., 2001*), their levels did not change proportionally upon starvation in LL, raising the question of whether the circadian oscillator function is intact under starvation conditions. We followed clock protein levels in constant darkness (DD), when the circadian clock displays a free running endogenous rhythm. FRQ protein showed a similar robust oscillation in both standard and starvation media, with no noticeable difference in period or phase (*Figure 1E*). When protein samples were analyzed on the same gel, increased FRQ phosphorylation was observed under starvation conditions in LL and at all time points in DD (*Figure 1F*). Similarly to the changes in LL, the expression of WC proteins was greatly reduced in DD upon glucose deprivation. However, neither phase nor amplitude of *frq* RNA oscillation was affected by starvation (*Figure 1G*, left panel), indicating that WCC activity was similar under both conditions. Since starvation does not affect *frq* RNA decay (see above), an unknown mechanism must recalibrate

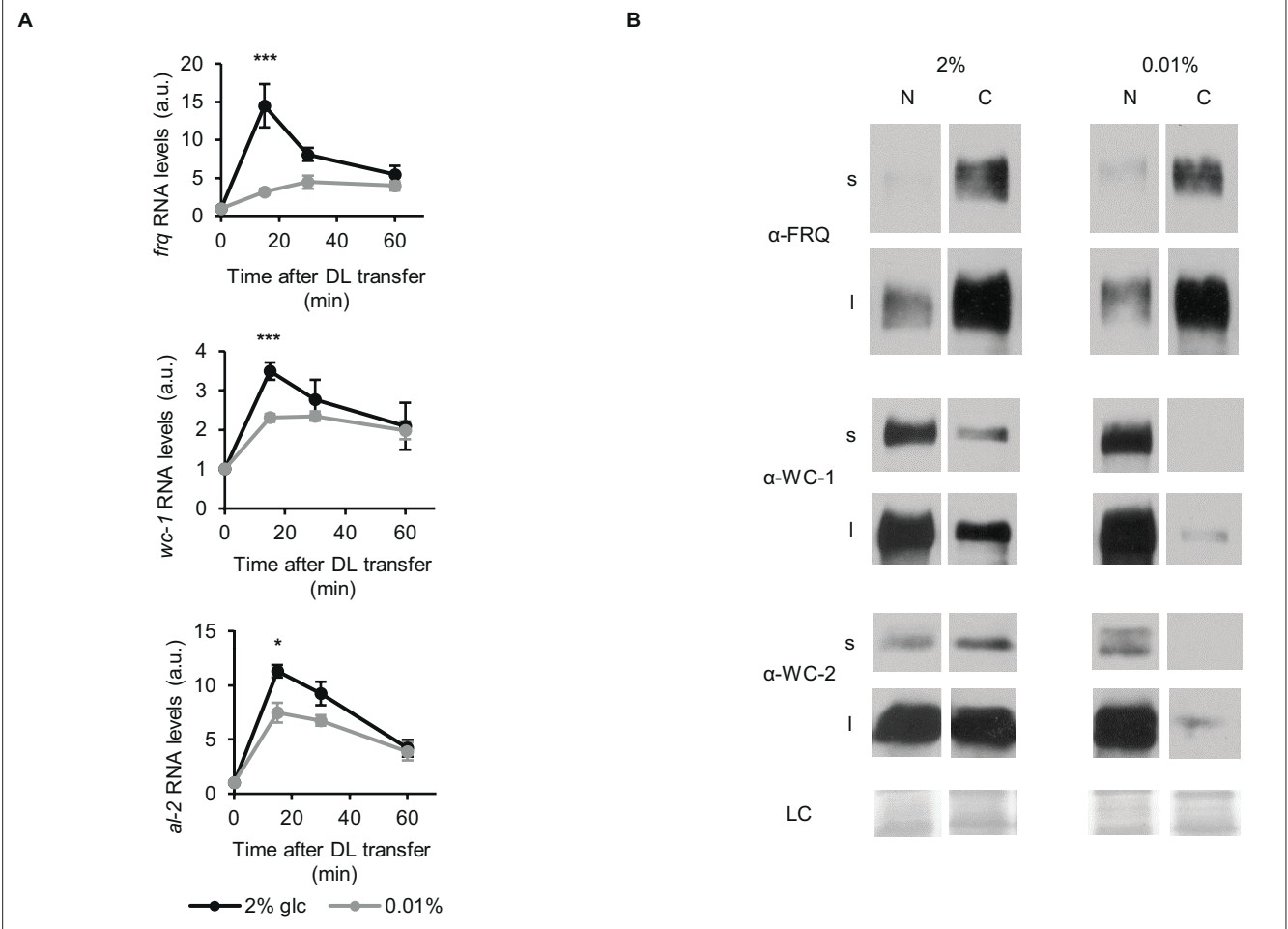

**Figure 2.** Glucose deprivation impacts both light induction of gene expression and subcellular distribution of clock components. (**A**) Light induction of gene expression is attenuated by glucose starvation. Mycelial discs of the *wt* strain were incubated in standard liquid medium for 24 hr, then transferred to media containing either 0.01% or 2% glucose. Following a 24 hr incubation in LL, cultures were transferred to DD for 16 hr and then light induced. Samples were harvested at the indicated time points after light on. Relative *frq*, *wc-1* and *al-2* RNA levels were normalized to that measured at the first time point. (n=5–11, ± SEM, Repeated measures ANOVA, significant time*treatment interaction, post hoc analysis: Tukey HSD test). (**B**) Glucose deprivation affects subcellular distribution of clock proteins. Growth conditions were as described in *Figure 1C*. Nuclear (**N**) and cytosolic (**C**) fractions were analyzed by Western blotting. (n=3, s: short exposure, l: long exposure, LC: loading control).

The online version of this article includes the following source data for figure 2:

**Source data 1.** Source data for *Figure 2B*.

the central clockwork to keep *frq* transcript levels and oscillation glucose-compensated despite the decline in WCC levels. To examine clock output function, we measured RNA levels of two clock-controlled genes, *ccg-2* and *fluffy* (*Bell-Pedersen et al., 1992*). Similarly to previous findings, starvation resulted in significant upregulation of *ccg-2* expression (*Bell-Pedersen et al., 1992*; *Kaldenhoff and Russo, 1993*; *Sokolovsky et al., 1992*). Interestingly, *fluffy* levels were also elevated under starvation compared to standard conditions. In addition, a robust oscillation of *ccg-2* and *fluffy* RNA was detected under both conditions, with peaks and troughs at the expected circadian time (*Figure 1G*, right panel; *Figure 1—figure supplement 2*). Our results suggest that the circadian clock functions robustly during glucose deprivation despite increased FRQ phosphorylation and decreased WCC levels and drives rhythmic expression of output genes without changes in period length or phase.

To examine the activity of the WCC as the photoreceptor of *Neurospora*, we followed the expression of the light-inducible genes *frq*, *wc-1* and *al-2* after dark-light (DL) transfer under both nutrient conditions. The light-induced initial increase of RNA levels was lower in starved than non-starved cultures, whereas the steady state expression levels after light adaptation were similar under both

conditions (*Figure 2A*). The difference in the kinetics of light induction suggests that the light-inducible pool or the photoreceptor function of WCC is reduced upon glucose deprivation.

Nucleocytoplasmic distribution of clock proteins is tightly associated with their phosphorylation and activity. Hence, we performed subcellular fractionation on our LL samples (*Figure 2B*). In accordance with previous data (*Cheng et al., 2005*; *Gyöngyösi et al., 2017*; *Schafmeier et al., 2006*), the majority of FRQ was in the cytosol fraction, and its distribution did not change markedly upon glucose-deprivation. In contrast, WC proteins were virtually absent from the cytosol of starved cells, whereas their nuclear concentrations were similar to those in the control cells.

## Multiple modulators are involved in the starvation response of the clock components

The objective of the following experiments was to explore possible mechanisms that could contribute to the glucose-dependent reorganisation of the TTFL components. Liquid cultures grown in LL were investigated. To address the role of FRQ-mediated feedback in this process, we used the FRQ-less mutant *frq⁹*. In *frq⁹* due to a premature stop codon, only a truncated, non-functional and unstable version of FRQ is expressed resulting in the loss of both the negative and positive feed-back of FRQ. One of the advantages of this strain is that activity of the *frq* promoter can be assessed by measuring *frq⁹* RNA levels in the absence of FRQ protein (*Aronson et al., 1994*; *Liu et al., 2019*). Increased amount of *frq⁹* RNA in standard medium compared with *frq* can be attributed to the absence of the negative feedback from FRQ to WCC (*Schafmeier et al., 2005*). However, while starvation did not alter *frq* expression in *wt*, it decreased the amount of *frq⁹* RNA (*Figure 3A*), resulting in similar RNA levels (reflecting similar *frq* promoter activity) in the two strains upon glucose withdrawal. The different behavior of *frq* promoter activity in *frq⁹* and *wt* in response to glucose suggests that FRQ-mediated processes contribute to the compensation of *frq* levels at different glucose availability. Amount of WC-1 did not change significantly in *frq⁹*, whereas WC-2 levels were moderately reduced in response to glucose depletion (*Figure 3B*). However, it is difficult to compare the glucose-dependence of WCC expression in *frq⁹* and *wt* because, due to the lack of the positive feedback of FRQ on the accumulation of the WCC, WC-1 and WC-2 levels are very low in *frq⁹* (*Cheng et al., 2001*; *Schafmeier et al., 2006*).

Action of CK1a on the clock proteins is dependent on its binding to FRQ via the FRQ-CK1a-interaction domain 1 (FCD1) (*Querfurth et al., 2011*) and FCD2 (*He et al., 2006*). CK1 contributes to the phosphorylation and subsequent inhibition of the WCC both in DD and LL. In addition, CK1a-driven phosphorylation of FRQ leads to its decreasing ability to exert the negative feedback, and also promotes phosphorylation-induced degradation of FRQ protein. To examine the glucose-dependent impact of CK1a on the circadian clock components, we used a *frqΔFCD1-2* strain. FRQΔFCD1-2 cannot recruit CK1a, and hence, the CK1a-dependent phosphorylation and inactivation of the WCC are impaired (*Querfurth et al., 2011*). Upon glucose withdrawal FRQΔFCD1-2 also displayed an electrophoretic mobility shift suggesting that starvation-induced phosphorylation of FRQ is not dependent on its interaction with CK1. The difference in WC levels between starved and non-starved cultures was moderate compared to *wt* (*Figure 3C*, upper panel). *frqΔFCD1-2* RNA levels were low under standard conditions and remained compensated upon glucose deprivation (*Figure 3C*, lower panel). Our data suggest that stable recruitment of CK1a to FRQ is not essential for the starvation-dependent hyperphosphorylation of FRQ and the compensation of *frq* RNA, but it slightly affects glucose-dependent changes of the WC levels.

Both FRQ and WC-1 can be modified by PKA (*Huang et al., 2007*; *Cha et al., 2008*). Moreover, PKA activity decreases in both *Neurospora* and yeast when glucose is limited (*Conrad et al., 2014*; *Li and Borkovich, 2006*). In the *mcb* strain, the regulatory subunit of PKA is defective and hence, PKA is constitutively active at an elevated level. In accordance with literature data, FRQ protein level was high in *mcb* compared to *wt* (*Huang et al., 2007*; *Figure 3D*). However, in the mutant glucose deprivation did not cause a reduction of WC-1 or WC-2 levels and *frq* RNA became elevated under starvation. This indicates that PKA plays a significant role in transducing starvation signals to the circadian clock.

Glycogen synthase kinase (GSK) is an important factor of the starvation response in many organisms including yeast (*Quan et al., 2015*) and was shown to fine-tune the circadian period in *Neurospora crassa* (*Tataroğlu et al., 2012*). As GSK is an essential protein, we used the *qa-gsk* strain in which GSK expression is under the control of the quinic acid (QA)-inducible *qa-2* promoter (*Tataroğlu et al.,*

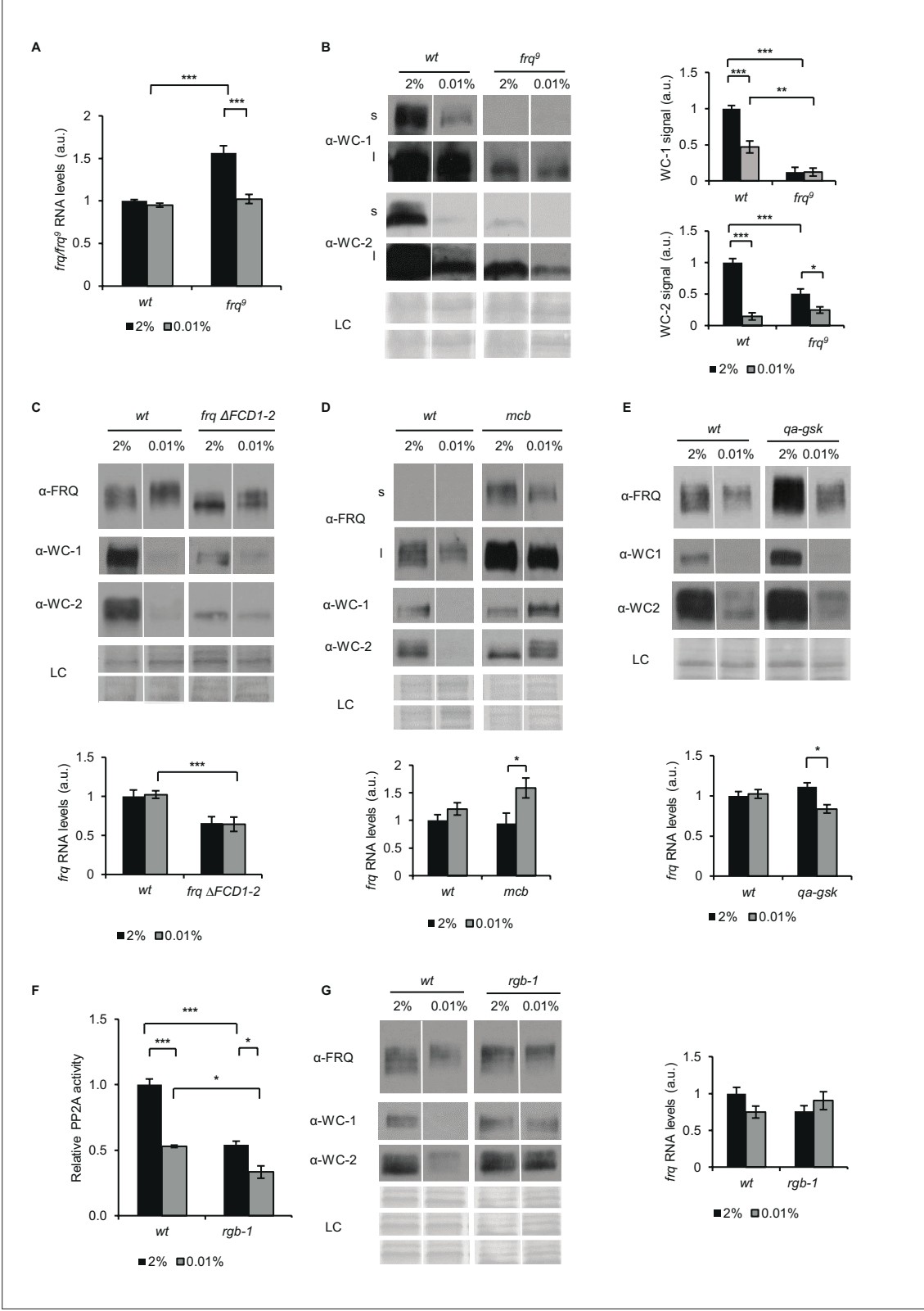

**Figure 3.** FRQ, PKA, GSK and PP2A affect the starvation response of the Neurospora clock. (**A**) *frq⁹* RNA expression is sensitive to glucose deprivation. Growth conditions were as described in ***Figure 1C***. RNA levels were normalized to that of *wt* grown in standard medium. (n=6, ± SEM, Factorial ANOVA; significant strain*treatment interaction; post hoc analysis: Tukey HSD test). (**B**) Effect of starvation on WC levels is reduced in *frq⁹*. Growth conditions were as described in ***Figure 1C***. Cell extracts were analyzed by western blotting (left panel). (n=3) Protein signal density was analyzed (right

*Figure 3 continued on next page*

*Figure 3 continued*

panel). (n=3, ± SEM, LC: loading control for WC-2 (upper panel) and WC-1 (lower panel), Factorial ANOVA; significant strain*treatment interaction; post hoc analysis: Tukey HSD test). (**C**) Impaired FRQ-CK1a interaction affects the starvation response of the molecular clock. Experiments were performed with the indicated strains as described in *Figure 1C*. Indicated protein (upper panel) and *frq* expressions (lower panel) were analyzed. RNA levels were normalized to that of *wt* grown in standard medium. (n (protein analysis)=12, LC: loading control for FRQ and WC-2 (upper panel) and WC-1 (lower panel), n (RNA analysis)=4–5, ± SEM, Factorial ANOVA; significant strain effect; post hoc analysis: Tukey Unequal N HSD test). (**D**) The starvation response is altered in the PKA mutant (*mcb*). Experiments were performed with the indicated strains as described in *Figure 1C*. Upper panel: analysis of cell extracts by Western blotting (n=12, s: short exposure, l: long exposure; LC: loading control for FRQ (upper panel), WC-1 and WC-2 (lower panel)) Lower panel: *frq* RNA levels of the indicated strains. RNA levels were normalized to that of *wt* grown in standard medium. (n=8–9, ± SEM, Factorial ANOVA; significant treatment effect; post hoc analysis: Tukey Unequal N HSD test). (**E**) Hyperphosphorylation of FRQ upon glucose withdrawal is dependent on GSK. Experiments were performed with the indicated strains as described in *Figure 1C*. The medium was supplemented with $1.5*10^{-5}$M quinic acid (QA) during the first day of incubation. Following the medium change, mycelia were incubated in QA-free medium. Upper panel: cell extracts analyzed by Western blotting. (LC: loading control) Lower panel: *frq* RNA levels of the indicated strains. RNA levels were normalized to that of *wt* grown in standard medium. (n=6, ± SEM; Factorial ANOVA, significant strain*treatment interaction, post hoc analysis: Tukey HSD test). (**F**) PP2A activity is decreased under starvation conditions. Experiments were performed with the indicated strains as described in *Figure 1C*. PP2A-specific activity of the cell lysates was determined and normalized to that of the *wt* grown in standard medium. (n=3–4, ± SEM, Factorial ANOVA, Significant strain*treatment interaction, post hoc analysis: Tukey Unequal N HSD test). (**G**) The starvation response is altered in the strain lacking a functional PP2A regulatory subunit (*rgb-1*). Experimental procedures were performed with the indicated strains as described in *Figure 1C*. Cell extracts were analyzed by Western blotting (n=12, LC: loading control for FRQ (upper panel), for WC-1 (middle panel) for WC-2 (lower panel)) (left panel) and RNA levels of *frq* were determined. RNA levels were normalized to that of *wt* grown in standard medium. (n=9–10, ± SEM, Factorial ANOVA, significant strain*treatment interaction) (right panel).

The online version of this article includes the following source data for figure 3:

**Source data 1.** Source data for *Figure 3B*.

**Source data 2.** Source data for *Figure 3C*.

**Source data 3.** Source data for *Figure 3D*.

**Source data 4.** Source data for *Figure 3E*.

**Source data 5.** Source data for *Figure 3G*.

*2012*). We performed our experiments without the addition of QA because under this condition the *qa-gsk* strain expresses GSK at very low level. According to previous findings (*Tataroğlu et al., 2012*), in the presence of 2% glucose WC-1 and FRQ levels were elevated in *qa-gsk* (*Figure 3E*). The starvation response of the clock was partially affected by GSK depletion (*Figure 3E*). While WC levels were reduced, even to a higher extent than in *wt*, FRQ did not become hyperphosphorylated and *frq* RNA levels moderately decreased upon starvation in the mutant. These results suggest that GSK supports the hyperphosphorylation of FRQ during starvation, but this modulation of FRQ is not sufficient to impact WCC levels.

PP2A is a major regulator of the clock affecting both FRQ and WCC (*Colot et al., 2006*; *Gyöngyösi and Káldi, 2014*; *Querfurth et al., 2011*; *Schafmeier et al., 2006*). Moreover, its activity is glucose-dependent in yeast (*Hughes Hallett et al., 2014*). As shown in *Figure 3F*, starvation decreases PP2A activity also in *Neurospora*, to levels characteristic for the *rgb-1* mutant, which lacks one of the regulatory subunits of the phosphatase complex. We have previously shown that dephosphorylation of WC-2 in CHX-treated cells depends on PP2A (*Gyöngyösi et al., 2013*; *Schafmeier et al., 2005*). Hence, the delayed dephosphorylation of WC-2 observed in the starved cells compared to the control cells (*Figure 1—figure supplement 1B*) can be a consequence of the decreased PP2A activity. Next, we examined the effect of glucose-starvation on the clock components in *rgb-1*. While *frq* RNA and FRQ protein levels were similar in *wt* and *rgb-1*, WC protein levels were not reduced in starved *rgb-1* (*Figure 3G*), indicating that PP2A affects the glucose-dependent change in the stoichiometry of clock components.

## Glucose-deprivation differentially impacts the transcriptome in *wt* and *Δwc-1*

Our results showed that despite the marked decrease of WC protein levels during starvation, WCC activity stayed preserved as reflected by *frq* RNA levels in both LL and DD. To examine the biological importance of maintaining WCC activity during starvation, we performed RNA-seq analysis in *wt* and *Δwc-1*. Liquid cultures were grown under standard and starving conditions in 12–12 hr dark-light

**Table 1.** The first 15 most upregulated annotated genes in *wt* during glucose starvation.
For experimental procedures of the RNA-seq analysis see Materials and Methods section. Genes with more pronounced upregulation in *wt* compared to *Δwc-1* are bold-lettered.

| ID | Name | Upregulation in *wt* (fold change) | Upregulation in *Δwc-1* (fold change) | *wt/Δwc-1* (2%) | Gene product |
|---|---|---|---|---|---|
| NCU02500 | *ccg-4* | **281** | n.s. | n.s. | **clock-controlled pheromone** |
| NCU07225 | *gh11-2* | **254** | 163 | n.s. | **xylanase** |
| NCU08769 | *con-6* | **117** | 88 | n.s. | **conidiation specific protein** |
| NCU05924 | *gh10-1* | **96** | 15 | n.s. | **xylanase** |
| NCU00943 | *tre-1* | 77 | 102 | n.s. | trehalase |
| NCU07325 | *con-10* | **70** | 13 | n.s. | **conidiation specific protein** |
| NCU08189 | *gh10-2* | **67** | 13 | n.s. | **xylanase** |
| NCU10055 | *nop-1* | 66 | 58 | n.s. | opsin |
| NCU06905 | *thnr* | **65** | n.s. | 0.01 | **tetrahydroxynaphthalene reductase** |
| NCU08457 | *eas* | 49 | 73 | n.s. | hydrophobin |
| NCU09873 | *acu-6* | **39** | 12 | 2.9 | **phosphoenolpyruvate carboxykinase** |
| NCU10021 | *hgt-1* | 35 | 56 | 1.4 | monosacharide transporter |
| NCU08755 | *gh3-3* | 28 | 38 | n.s. | beta-glucosidase |
| NCU00762 | *gh5-1* | **21** | n.s. | n.s. | **endoglucanase** |
| NCU08114 | *cdt-2* | **21** | 11 | n.s. | **hexose transporter** |

cycles and samples were harvested after 48 hr (ZT 12, end of the light period). Under these conditions, the clock is entrained in *wt*, and at ZT 12 an adapted state for light-dependent genes can be expected as it was shown for *frq* RNA (*Elvin et al., 2005*). Long-term glucose starvation affected more than 20% of coding genes at the level of RNA. The most upregulated genes in *wt* encode polysaccharide degrading enzymes, conidiation specific proteins and monosaccharide transporters, similarly to findings in *Aspergillus niger* (*Nitsche et al., 2012*). Polysaccharide degrading enzymes (xylanases, endoglucanase) are considered scouting factors and their induction might play a foraging role increasing survival chances under starvation conditions (*Benocci et al., 2017*). Eleven out of the 15 genes that were the most upregulated in *wt* responded with a weaker or no increase to glucose-deprivation in *Δwc*-1, indicating a limited transcriptomic adaptation in the absence of the WCC (*Table 1*). *Figure 4A and B* show the number and behavior of genes that responded differently to glucose withdrawal in *wt* and *Δwc*-1 or showed differences in their *Δwc-1/wt* expression ratios between starved and non-starved conditions, with a twofold significant difference as a threshold. In a further analysis, we focused on physiologically more relevant differences and found that a sum of 1377 RNAs representing 13% of the 9758 coding genes changed in a strain-specific manner in response to glucose starvation (*Figure 4—figure supplement 1*). Based on literature data showing that the WCC affects the expression of several other transcription factors and controls basic cellular functions which might affect the expression of further genes, it was not surprising that only 90 out of the 1377 genes were reported to be direct targets of the WCC (*Smith et al., 2010*; *Hurley et al., 2014*; *Figure 4—source data 1*). Detailed comparison of the *Δwc-1/wt* ratios of transcript levels revealed that 1348 genes displayed glucose-dependent difference showing that effect of *wc-1* mutation on the transcriptome is highly nutrient-dependent (*Figure 4—figure supplement 2*).

Using the Gene Ontology (GO) enrichment tool (*Mi et al., 2013*), we found that a substantial proportion of the genes responding differently to starvation in *wt* and *Δwc*-1 is involved in crucial metabolic processes (*Figure 4C and D*, *Figure 4—figure supplement 3*, *Figure 4—source data 2*). Further analysis showed that in the *wt* the downregulation of biosynthetic processes of small molecules and

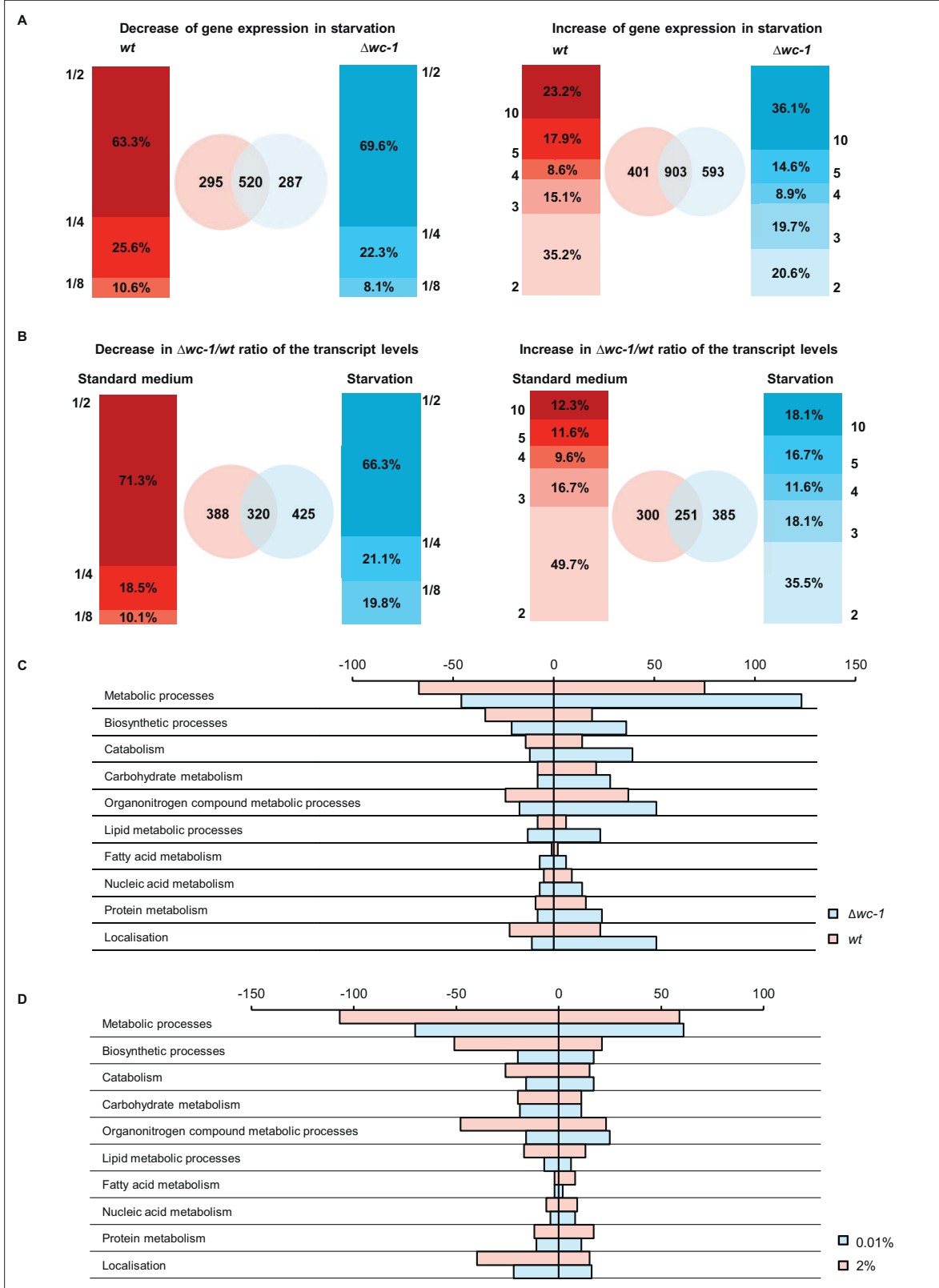

**Figure 4.** WC-1 is required for adaptation to starvation in genome-wide scale. (**A**) Distribution of number of genes showing starvation induced up- and downregulation in *wt* and *Δwc-1*. Values on the y-axis of the bar graphs indicate the minimal and maximal fold-change of up- and downregulation, respectively. Venn-diagrams indicate the number of up- and downregulated genes in *wt* (red) and *Δwc-1*(blue). (**B**) Distribution of genes expressed at lower and higher level in *Δwc-1* than in *wt* in standard or starvation medium. Values on the y-axis of the bar graphs indicate the minimal and maximal

*Figure 4 continued on next page*

*Figure 4 continued*

ratios of RNA levels (Δwc-1/wt). Venn-diagrams indicate the number of genes showing different expression in the two strains in the indicated medium (red: standard medium, blue: starvation). (**C**) Number of genes showing strain-specific changes in major metabolic functions in response to a 40 hr glucose deprivation. Positive and negative values indicate number of genes with increased and decreased RNA levels, respectively. Genes were classified by GO analysis (**Mi et al., 2013**). (**D**) Number of genes showing treatment-specific (2% vs 0.01% glucose) changes in their Δwc-1/wt RNA ratio. Positive and negative values indicate number of genes with increased and decreased RNA ratio, respectively. Genes involved in major metabolic functions were classified by GO analysis (**Mi et al., 2013**).

The online version of this article includes the following source data and figure supplement(s) for figure 4:

**Source data 1.** Genes, that changed in a strain-specific manner in response to glucose starvation and are direct targets of the WCC.

**Source data 2.** Gene Ontology (GO) enrichment analysis of genes showing at least two-fold significant alteration in their amount in response to starvation.

**Source data 3.** Gene Ontology (GO) enrichment analysis of genes showing strain-specific response to starvation.

**Source data 4.** Genes of central carbon metabolism, amino acid biosynthesis and fatty acid metabolism, that showed strain-specific expression change to starvation.

**Source data 5.** Comparison of the results from RNA-seq and the experimental validation of the chosen genes with qPCR.

**Figure supplement 1.** Strain-specific differences of gene expression changes in response to starvation.

**Figure supplement 2.** Glucose-specific differences in Δwc-1/wt ratio of gene expression.

**Figure supplement 3.** Characterization of different metabolic pathways in strain-specific responses to glucose starvation.

**Figure supplement 4.** *Genes of central carbon metabolism, that showed strain-specific change to starvation.*

**Figure supplement 5.** *Genes of amino acid biosynthesis, that showed strain-specific change to starvation.*

**Figure supplement 6.** *Genes of fatty acid metabolism, that showed strain-specific change to starvation.*

**Figure supplement 7.** *qPCR validation of the RNA-seq data.*

amino acids dominated, whereas Δwc-1 had a higher tendency to up-regulate amino acid catabolism and down-regulate fatty acid synthesis in response to long-term starvation (**Figure 4—source data 3**). In analysis using the KEGG database, strain-specific differences in genes assigned to glycolysis, citrate and glyoxylate cycle, the pentose phosphate pathway and amino acid metabolism were identified (**Figure 4—figure supplements 4–6**). However, pathways of fatty acid biosynthesis displayed the most uniform strain-dependent differences in response to starvation with a dominant downregulation in Δwc-1 compared to *wt* (**Figure 4—figure supplement 6**, **Figure 4—source data 4**). Among constituents of these pathways only tetrahydroxynaphthalene reductase-2 was strain-specifically upregulated in *wt* (**Table 1**). Importantly, this enzyme catalyzes key steps of the synthesis of melanin, which is a crucial factor required for mechanical strength of cell wall and is therefore central in the adaptation to extreme changes in the environmental conditions (**Nosanchuk et al., 2015**).

To validate RNA-seq data, we performed qRT-PCR measurements for genes involved in the metabolism of carbohydrates or amino acids as well as for genes important for conidiation, the well-characterized nutrient-dependent process in fungi (**Springer, 1993**; **Figure 4—figure supplement 7**). All tested RNAs showed similar expression ratios in both analyses indicating the high reliability of our transcriptome profiling data (**Figure 4—source data 5**). Striking differences between Δwc-1 and *wt* were observed for the conidiation-specific protein 10 (*con-10*, NCU07325), the conserved mitochondrial gene *tca-3* (NCU02366), encoding a putative aconitase that catalyzes the citrate-isocitrate transition, and the gene of choline dehydrogenase (*choldh*, NCU01853) promoting the production of glycin-betain from choline, and thereby participating in amino acid metabolism.

## Efficient recovery from starvation is dependent on a functional clock

Based on the transcriptomic data indicating significant differences between *wt* and Δwc-1 in the adaptation to glucose limitation, we hypothesized that starved *wt* might be better prepared to growth regeneration and could respond more effectively to the resupply of glucose than clock deficient strains. *wt* and the clock mutants *frq*[10] and Δwc-1 were incubated in light-dark cycles and following the 40 hr starvation the medium was supplemented with 2% glucose (**Figure 5A**). Interestingly, the medium of starved *wt* cultures was cloudy before the addition of glucose, whereas the medium of clock-less cultures remained clear (**Figure 5B**, Before: upper panel). As observed after filtration of the cultures (**Figure 5B**, Before: lower panel), the cloudy material was constituted of small satellite

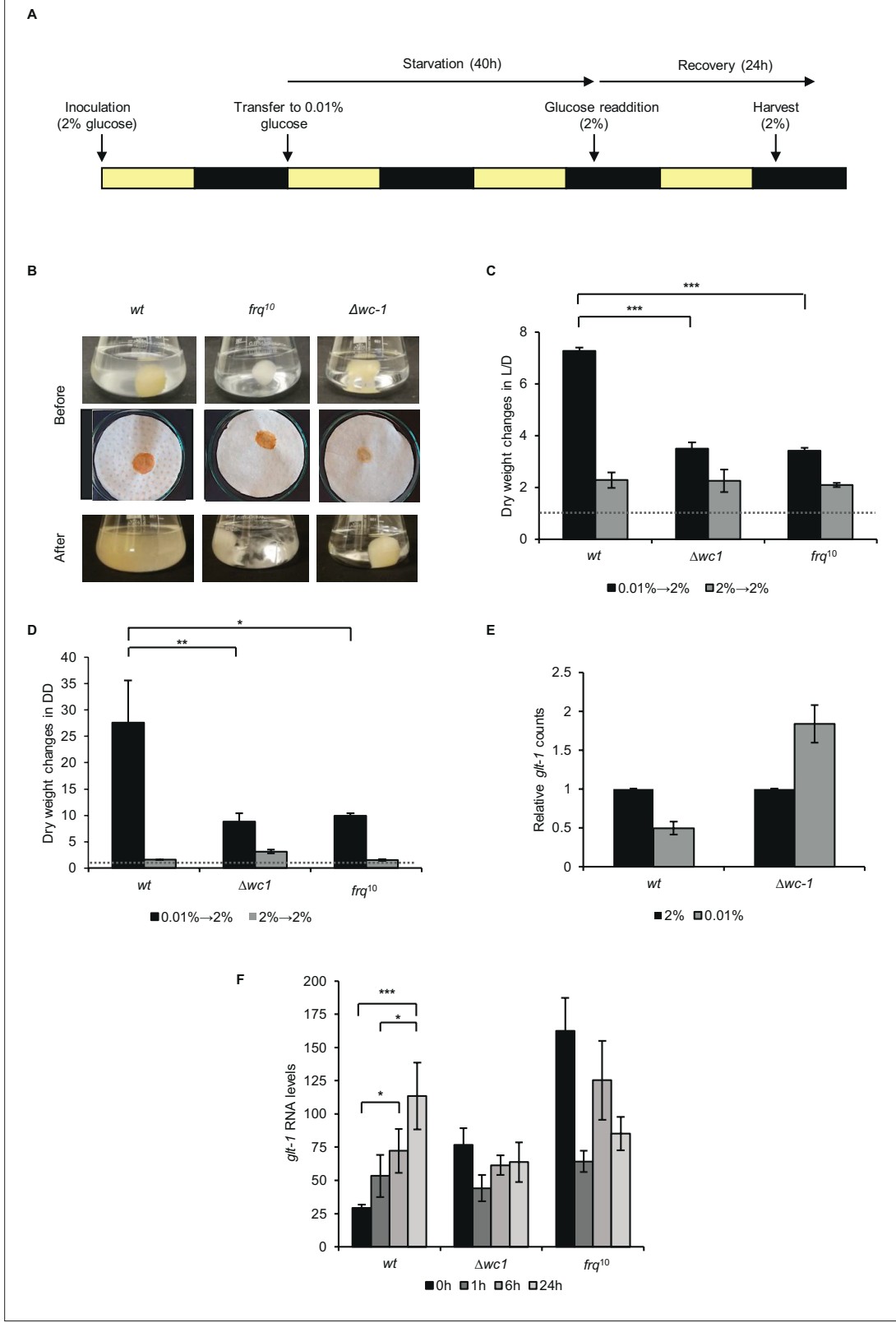

**Figure 5.** *Proper recovery from starvation requires a functional clock.* (**A**) Schematic design of the experiment. Mycelial discs of the *wt* and the clock mutant strains were incubated in standard liquid medium in L/D12. After 24 hr mycelial balls were transferred to starvation medium. Following 40 hr of starvation, glucose was added to the medium. Yellow and black bars indicate the periods cultures spent in light and darkness, respectively. (**B**) Comparison of the growth of *wt* and clock mutants after glucose resupply. Pictures of the liquid (Before: upper images) and the vacuum filtered

*Figure 5 continued on next page*

*Figure 5 continued*

cultures (Before: lower images) were taken after 40 hr of starvation and 24 hr after glucose resupply (After). (**C**) *wt* grows faster after glucose supplementation than clock mutants in L/D. Experimental procedures were performed as described in (**A**). Dry weight of cultures was measured after 24 hr of glucose resupply. Values were normalised to the dry weight measured before glucose resupply (indicated with dashed line). (n=4, ± SEM, Factorial ANOVA, significant strain*treatment interaction, post hoc analysis: Tukey HSD test). (**D**) *wt* grows faster after glucose supplementation than clock mutants in DD. Experimental procedures were performed as described in (**A**) except the light conditions: after 24 hr in LL, cultures were incubated in DD. Values were normalised to the dry weight measured before glucose resupply (indicated with dashed line). (n=3, ± SEM, Factorial ANOVA, significant strain*treatment interaction, post hoc analysis: Fisher LSD test). (**E**) Lack of *wc-1* affects the proper glucose transporter expression in response to starvation. *glt-1* counts in RNAseq data. Values were normalized to that of cultures grown in 2% glucose. (**F**) Lack of the functional clock affects the proper alignment of glucose transporter expression to glucose levels. Experiments were performed as described in (**A**). Samples were harvested at the indicated time points following glucose readdition and relative levels of *glt-1* RNA were determined by qPCR. (n=3, ± SEM, Factorial ANOVA, significant strain*treatment interaction, post hoc analysis: Tukey HSD test).

The online version of this article includes the following figure supplement(s) for figure 5:

**Figure supplement 1.** *Growth rate of wt recovers faster, than that of Δwc-1 and frq^{10} on solid medium.*

colonies which might serve as a source for growth reinitiation. Accordingly, the growth rate reflected by dry weight increase upon glucose resupply was higher in *wt* cultures than in *frq*$^{10}$ and *Δwc-1*, whereas non-starved control samples did not show strain-specific differences in the acceleration of growth after the transfer into fresh glucose-rich medium (*Figure 5B*, After, 5 C). To separate the clock and photoreceptor functions of the WCC in growth recovery, we repeated the experiment with cultures incubated in DD, that is when the WCC is not active as photoreceptor (*Figure 5D*). Growth of both *frq*$^{10}$ and *Δwc-1* was significantly slower than that of *wt*, suggesting that presence of a functional clock supports regeneration upon glucose addition. Recovery from starvation was tested also on solid media by comparing growth rates following the transfer of mycelia from starved and control liquid cultures onto race tube medium which contained glucose (*Figure 5—figure supplement 1*). Growth rates were determined for three consecutive periods. While the growth rate of starved *wt* cultures approached that of the non-starved control during the second 6-hr period, it remained at a low level in *Δwc-1*, and slightly but significantly lagged behind the control rate also in *frq*$^{10}$.

In *Neurospora*, a dual-affinity transport system mechanism enables the proper adaptation of glucose uptake to external glucose levels. When glucose levels are elevated, increased expression of the low-affinity and high-capacity transporter, *glt-1* ensures efficient utilization of the carbon source (*Wang et al., 2017*). Accordingly, the transcriptomic analysis revealed higher *glt-1* levels in non-starved than starved *wt* cells. In *Δwc-1*, however, *glt-1* was upregulated under starvation conditions, indicating that the WCC is crucial in the adaptation of glucose transport to nutrient levels (*Figure 5E*). Next, we examined *glt-1* expression during recovery. In *wt*, *glt-1* RNA levels gradually increased upon supplementation of the starvation medium with 2% glucose. In contrast, expression of *glt-1* was relatively high in *Δwc-1* and *frq*$^{10}$ under starvation, whereas it decreased or showed no change upon glucose addition, indicating that adaptation of the transporter level to glucose availability is disturbed in the clock mutants (*Figure 5F*). In summary, the marked differences between the recovery behavior of the different strains suggest that adaptation to changing nutrient availability is more efficient when a circadian clock operates in the cell.

## Discussion

In nature, the reproductive capacity of living things exceeds the availability of resources, and organisms face nutrient shortages from time to time. Optimal metabolic responses to nutrient deficiencies can therefore provide a significant advantage in selection. Deprivation of carbon sources reduces the transcription/translation rate, which is also a challenge for the TTFL-based molecular clock. Nutrient compensation of the circadian rhythm and proper timely regulation of physiology could therefore be a prerequisite for appropriate adaptation. Short glucose deprivation does not affect expression of the core clock proteins in *Neurospora* and this maintenance of the protein levels was considered essential for nutrient-compensation (*Emerson et al., 2015*; *Gyöngyösi et al., 2017*; *Sancar et al., 2012*). Our experiments show that the circadian TTFL functions with substantially altered levels and stoichiometry of its key elements during long-term glucose starvation. Thus, WCC levels decrease and FRQ becomes hyperphosphorylated, while the clock maintains the period of expression of *frq* and

various output genes. Importantly, cytosolic but not nuclear WC protein levels are primarily reduced under low glucose conditions. Cytosolic WCC might serve as a reserve pool of the positive factor that can be rapidly activated by Zeitgebers (e.g. light) to synchronize the clock with the changing environment (*Malzahn et al., 2010*). In accordance with this, we found that acute light response is reduced in glucose-starved cells. However, the maintained nuclear levels of the WCC might contribute to the preserved function of the circadian clock under starvation.

Our results indicate that presence of a functional circadian clock significantly affects the adaptation of *Neurospora crassa* to long-term glucose deprivation. Starved *wt* cultures better adapt the expression of the high-capacity glucose transporter to glucose levels and grow faster in both LD and DD upon resupply of glucose than clock-less cultures. Although WCC level decreases during starvation, *frq* transcription is compensated, indicating that the reduced WCC pool is still able to exert the same activity. Accordingly, we found that more than 1300 coding genes displayed different response to glucose starvation in *wt* and *Δwc-1*. These genes affect various cell functions including carbohydrate, amino acid and fatty acid metabolism. The lack of WCC resulted in a characteristic shift from the control of central carbon metabolism and the downregulation of amino acid synthesis to the upregulation of amino acid catabolism and the downregulation of fatty acid synthesis. This means that the WCC is an important determinant of the adaptive reorganization of the transcriptome in response to critical changes in glucose supply. Further investigations could differentiate between the clock and photoreceptor functions of the WCC in the glucose-dependent control of the transcriptome.

Glucose availability was shown to act as an effective Zeitgeber at different levels of the phylogenetic hierarchy, with detailed investigations in *Arabidopsis thaliana*, *Drosophila melanogaster* and mammalian systems (*Frank et al., 2018*; *Kaasik et al., 2013*; *Hirota et al., 2002*). In *Drosophila* starvation resistance is reduced in the clock mutants *per*[01] and *tim*[01](*Kaasik et al., 2013*; *Schäbler et al., 2020*), also indicating that endogenous time measuring helps to adapt to critical metabolic conditions. In mammal, glucose withdrawal cannot be examined at the organismal level because blood glucose concentrations below a critical threshold (3.2 mmol/l in human) can lead to life-threatening conditions. Nevertheless, severe glucose depletion may occur locally when blood supply of the tissue is impaired and consequences of this condition can be examined in cell culture models. In mouse fibroblasts normal amplitude cycling of PER2 was detected for several days upon glucose depletion, indicating that, similarly to our findings in *Neurospora*, the clock function is preserved during critical metabolic conditions in mammalian cells as well (*Putker et al., 2018*). However, still no data are available about the effect of starvation on the molecular network of the clock in any other organisms than *Neurospora*.

In conclusion, our results suggest that the WCC has a major impact on the mechanisms balancing the cellular energy state in *Neurospora* and highlight the importance of the circadian clock in cell survival under nutritional stress.

## Ideas and speculation

Based on our findings the question arises what cellular mechanisms might lead to the characteristic reorganization of the molecular clock during long-term glucose deprivation and how the reduced levels of the WCC can maintain normal transcriptional activity, at least at the level of *frq* RNA. As RNA levels encoding the WCC subunits remain compensated during glucose starvation, posttranscriptional mechanisms may be involved in the reduction of their protein levels. A possible explanation is that hyperphosphorylation of FRQ observed in starvation weakens the inhibitory feedback on the WCC which might contribute to the compensation for the reduced transcription/translation rate during nutrient shortage but also leads to faster degradation of the WCC (*Figure 6*). The starvation-dependent reduced stability of WC-1 in *wt* supports this concept. The absence of FRQ-dependent feedback on the WCC might explain why WCC levels are only moderately affected by starvation in *frq*[9]. On the other hand, in *frq*[9] WCC levels are low even under standard glucose conditions, raising the possibility that starvation can not decrease it efficiently, that is, below a minimum level. However, when analysed from another perspective, elevation of the glucose level seems to increase the amount of the WCC in a FRQ-dependent manner. The starvation-dependent decrease in *frq*[9] RNA abundance also suggests that in the absence of the feedback mechanism the starvation-induced reduction of the global transcription rate cannot be compensated by an increase of the WCC activity. Our findings reveal that the earlier described difference in the *frq* promoter activity between *wt* and *frq*[9] (*Schafmeier et al., 2005*) is absent under starvation conditions, that is, the phenotype of the mutant is

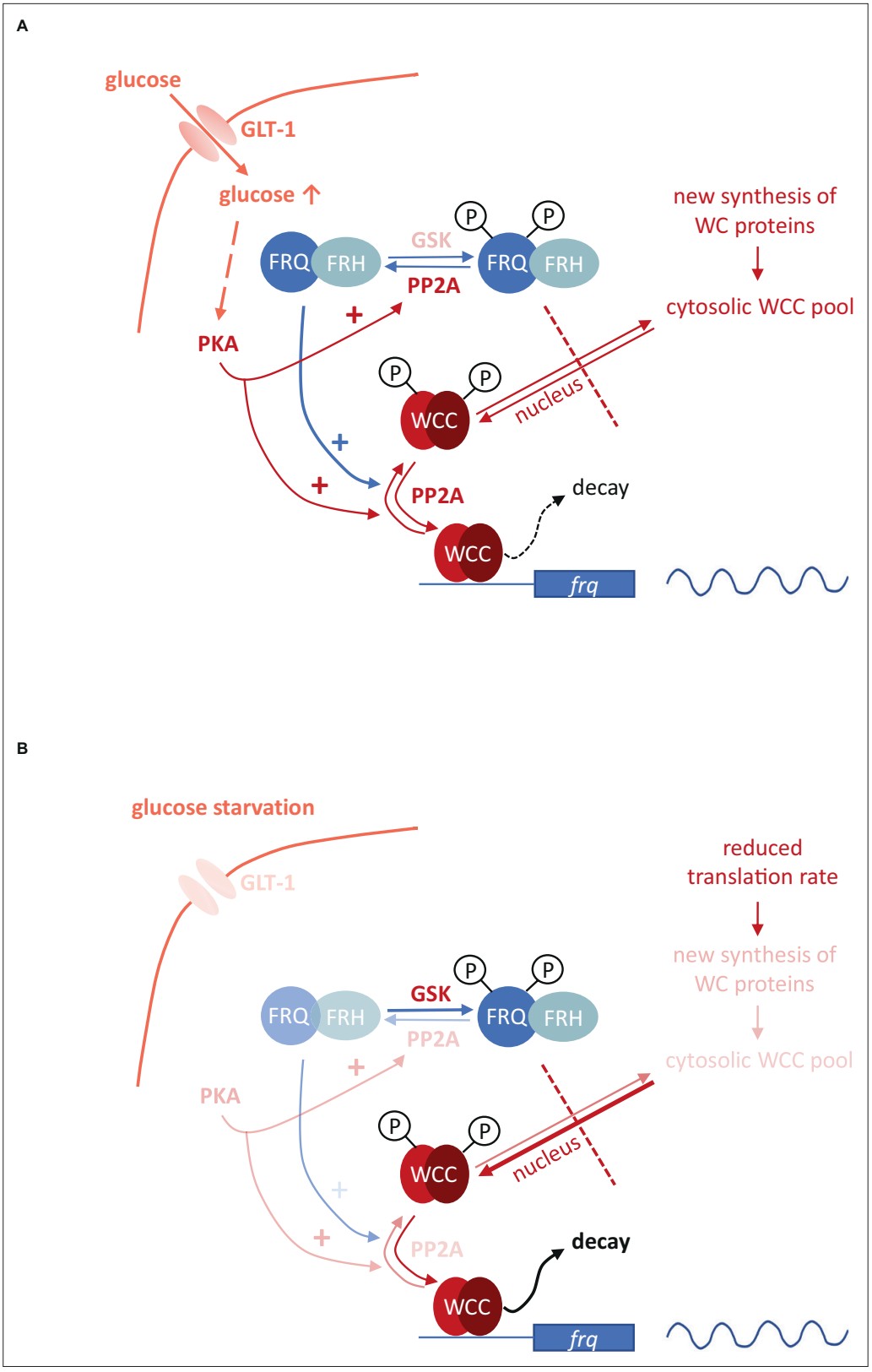

**Figure 6.** *Model representing the role of the negative feedback and the PKA-, PP2A- and GSK-mediated signaling in the control of the molecular clock at high glucose levels (**A**) and under starvation (**B**). Starvation reduces the activity of both PKA and PP2A but stimulates GSK. PKA can act as a central regulator of the starvation-induced modifications of the clock components, as its weakened activity results in enhanced action and the consequent*

*Figure 6 continued on next page*

*Figure 6 continued*

destabilization of the WCC, resulting in compensated *frq* transcription at significantly reduced WC levels. PKA also affects PP2A. Reduced activity of PP2A and the parallel induction of GSK in starvation can lead to hyperphosphorylation of FRQ which in turn lessens the negative feedback on the WCC. Higher and lower activities of enzymes and processes are indicated by more and less intense colors, respectively.

nutrient-dependent. The glucose-dependent modulation of the clock is probably not restricted to changes in the FRQ-dependent recruitment of CK1a to the WCC, as the impaired interaction between FRQ and CK1a in the *frqΔFCD1-2* strain only partially affected WCC level alterations and did not change *frq* RNA level. In addition, the starvation-induced phosphorylation of FRQ was preserved in the mutant suggesting that CK-1 independent mechanisms might be responsible for it.

Starvation triggers characteristic changes in the activity of signaling routes that affect basic components of the circadian clock. Although the multifunctional pathways might act via pleiotropic mechanisms as well, based on their earlier characterized role in the control of the *Neurospora* clock, their action can be inserted into a model describing the glucose-dependent reorganization of the oscillator (*Figure 6*). We found that PP2A, GSK and PKA are involved in the starvation-induced modification of the molecular clock. PP2A was described as glucose sensitive regulator in fungal as well as in mammalian cells (*Hughes Hallett et al., 2014*; *Lee et al., 2018*) and we showed that glucose starvation decreases PP2A activity in *Neurospora*. PP2A is known to act on both the negative and the positive element of the *Neurospora* clock (*Schafmeier et al., 2005*; *Yang et al., 2004*). Reduced PP2A activity during nutrient limitation may contribute to the hyperphosphorylation of FRQ which lessens the negative feedback on the WCC activity. However, this activator effect might be antagonized by a weaker dephosphorylation of the WC proteins. In the *rgb-1* strain in which one of the regulatory subunits of PP2A is disrupted and the enzyme activity is rather low, WC levels did not respond to glucose-deprivation, suggesting that starvation-dependent expression change of the WCC is dependent on PP2A. Starvation triggers activation of GSK in many organisms. Our data indicate that hyperphosphorylation of FRQ upon glucose depletion is dependent on GSK. Whether FRQ is a direct substrate of GSK, is still to be elucidated. Reduced PKA activity in starvation is considered as a central organizer of the cellular adaptation to nutrient restriction (*Huang et al., 2007*; *Li and Borkovich, 2006*). PKA is a priming kinase for subsequent phosphorylation of the WCC by CK1a and/or CK2, contributing to inactivation of the transcription factor (*Huang et al., 2007*). Thus, in addition to the reduced feedback by FRQ, the weakened inhibition by PKA might also ensure that transcriptional activity of the WCC is preserved during starvation in both DD and LL. However, since PKA increases FRQ stability (*Huang et al., 2007*), affects translation under stress conditions (*Barraza et al., 2017*; *Leipheimer et al., 2019*) and activates PP2A in yeast (*Castermans et al., 2012*), it may also influence the clock via these functions. Indeed, the enhanced PKA activity in the *mcb* strain interfered with all observed starvation-induced changes within the molecular oscillator. In conclusion, our observations along with literature data suggest a model in which PKA acts as a main coordinator of the adaptation of the circadian clock to different nutrient conditions (*Figure 6*). Overall, the nutrient-dependent activity of multiple signaling pathways might ensure that the transcriptional activity of the WCC and thus expression and oscillation of *frq* remain nutrient-compensated.

## Materials and methods

### Key resources table

| Reagent type (species) or resource | Designation | Source or reference | Identifiers | Additional information |
|---|---|---|---|---|
| Strain, strain background (*Neurospora crassa*) | *wt* | Fungal Genetics Stock Center | #2489 | |
| Strain, strain background (*Neurospora crassa*) | *wt,bd* | Fungal Genetics Stock Center | #1858 | |
| Strain, strain background (*Neurospora crassa*) | *bd;frq10* | Fungal Genetics Stock Center | #7490 | |

*Continued on next page*

*Continued*

| Reagent type (species) or resource | Designation | Source or reference | Identifiers | Additional information |
|---|---|---|---|---|
| Strain, strain background (*Neurospora crassa*) | *bd;frq9* | Fungal Genetics Stock Center | #7779 | |
| Strain, strain background (*Neurospora crassa*) | *rgb-1* | Fungal Genetics Stock Center | #8380 | |
| Strain, strain background (*Neurospora crassa*) | *mcb* | Fungal Genetics Stock Center | #7094 | |
| Strain, strain background (*Neurospora crassa*) | *bd; Δwc1* | https://doi.org/10.1093/emboj/20.3.307 https://doi.org/10.1093%2Femboj%2F18.18.4961 | | |
| Strain, strain background (*Neurospora crassa*) | *bd;frq10, his-3* | https://doi.org/10.1128%2Fmcb.16.2.513 | | |
| Strain, strain background (*Neurospora crassa*) | *frq10 Δfcd1-2* | This paper | | See Materials and Methods. |
| Strain, strain background (*Neurospora crassa*) | *qa-gsk* | https://doi.org/10.1074%2Fjbc.M112.396622 | | |
| Strain, strain background (*Escherichia coli*) | *ΔH5-α* | New England Biolabs | | |
| Antibody | α-FRQ (mouse monoclonal) | https://doi.org/10.1093/emboj/20.24.7074 | | WB (1:5000) |
| Antibody | α-WC1 (rabbit polyclonal) | https://doi.org/10.1101%2Fgad.360906 | | WB (1:10000) |
| Antibody | α-WC2 (rabbit polyclonal) | https://doi.org/10.1038%2Fembor.2008.113 | | WB (1:10000) |
| Antibody | Goat- α-mouse IgG (H/L): HRP, polyclonal | Bio-Rad | Cat#1706516 | WB (1:5000) |
| Antibody | Goat α-rabbit IgG (H/L): HRP, polyclonal | Bio-Rad | Cat#1706515 | WB (1:5000) |
| Recombinant DNA reagent | *pBM60-ClaI-ΔFCD1-2* (plasmid) | https://doi.org/10.1016/j.molcel.2011.06.033 | | |
| Sequence-based reagent | *frq F* | https://doi.org/10.1093/emboj/20.24.7074 | qPCR primer | TTGTAAT GAAAGGT GTCCGAA GGT |
| Sequence-based reagent | *frq F* | https://doi.org/10.1093/emboj/20.24.7074 | qPCR primer | GGAGGAA GAAGCGG AAAACA |
| Sequence-based reagent | *frq probe* | https://doi.org/10.1093/emboj/20.24.7074 | qPCR primer | [6-FAM] AC CTCCCAAT CTCCGAAC TCGCCTG [TAMRA] |
| Sequence-based reagent | *wc-1 F* | https://doi.org/10.1101%2Fgad.360906 | qPCR primer | ACCTCGCT GTCCTCGA TTTG |
| Sequence-based reagent | *wc-1 R* | https://doi.org/10.1101%2Fgad.360906 | qPCR primer | TGCTGGGC CTCTTTCAA CTC |
| Sequence-based reagent | *wc-1 probe* | https://doi.org/10.1101%2Fgad.360906 | qPCR primer | [6-FAM] CC GTCCGAC ATCGTGC CGG [TAMRA] |

*Continued on next page*

*Continued*

| Reagent type (species) or resource | Designation | Source or reference | Identifiers | Additional information |
|---|---|---|---|---|
| Sequence-based reagent | *wc-2 F* | https://doi.org/10.1038%2Fembor.2008.113 | qPCR primer | AGTTTGCA CCCAATCC AGAGA |
| Sequence-based reagent | *wc-2 R* | https://doi.org/10.1038%2Fembor.2008.113 | qPCR primer | AGGGTCG AAGCCAT CATGAAC |
| Sequence-based reagent | *wc-2 probe* | https://doi.org/10.1038%2Fembor.2008.113 | qPCR primer | [6-FAM] AG TCGCCTTT CTGCCAG [TAMRA] |
| Sequence-based reagent | *ccg-2 F* | This paper | qPCR primer | GCTGCGT TGTCGGT GTCAT |
| Sequence-based reagent | *ccg-2 R* | This paper | qPCR primer | GGAGTTG CCGGTGT TGGTAA |
| Sequence-based reagent | *ccg-2 probe* | This paper | qPCR primer | [6-FAM] AA TGTGGTG CCAGCGT CAAGTGC TG [TAMRA] |
| Sequence-based reagent | *al-2 F* | https://doi.org/10.1016/j.cell.2010.08.010 | qPCR primer | ACCTGGC CAATTCG CTCTTT |
| Sequence-based reagent | *al-2 R* | https://doi.org/10.1016/j.cell.2010.08.010 | qPCR primer | GACAGAA GGAGTAC AGCAGGA TCA |
| Sequence-based reagent | *al-2 probe* | https://doi.org/10.1016/j.cell.2010.08.010 | qPCR primer | [6-FAM] CT GGTCGAC TCCGCAT T [TAMRA] |
| Sequence-based reagent | *act F* | https://doi.org/10.1101%2Fgad.360906 | qPCR primer | AATGGGT CGGGTAT GTGCAA |
| Sequence-based reagent | *act R* | https://doi.org/10.1101%2Fgad.360906 | qPCR primer | CTTCTGG CCCATAC CGATCA |
| Sequence-based reagent | *act probe* | https://doi.org/10.1101%2Fgad.360906 | qPCR primer | [6-FAM] CA GAGCTGT TTTCCCT TCCATCG TTGGT [TAMRA] |
| Sequence-based reagent | *gna-3 F* | This paper | qPCR primer | ATATCCT CACTTGA CACAAGC C |
| Sequence-based reagent | *gna-3 R* | This paper | qPCR primer | CGGAGTC TTTAAGG GCGTTAT T |
| Sequence-based reagent | *gna-3 probe* | This paper | qPCR primer | [6-FAM] TC CAACATC CGTCTCG TGTTTGC T [TAMRA] |

Continued

| Reagent type (species) or resource | Designation | Source or reference | Identifiers | Additional information |
|---|---|---|---|---|
| Sequence-based reagent | tfc-1 F | This paper | qPCR primer | CGATTTG ATCCCTC CTCCTAA C |
| Sequence-based reagent | tfc-1 R | This paper | qPCR primer | GGGCTGA TTTCCTT GGTGTA |
| Sequence-based reagent | tfc-1 probe | This paper | qPCR primer | [6-FAM] AT GAGCTTG CCCTTCC AATACGG T[TAMRA] |
| Sequence-based reagent | sarA F | This paper | qPCR primer | TGGTTGT GGTCTTG GTTCTAC |
| Sequence-based reagent | sarA R | This paper | qPCR primer | TGGCAAC GCGATCA TTCT |
| Sequence-based reagent | sarA probe | This paper | qPCR primer | [6-FAM] AT ATCCTTT CCAACCT CGGCCTG C[TAMRA] |
| Sequence-based reagent | aga-1 F | This paper | qPCR primer | CAGTGTC AAGAAGC TGGTCTA C |
| Sequence-based reagent | aga-1 R | This paper | qPCR primer | TGCCGTG CTTGTCA ATGT |
| Sequence-based reagent | gln-1 F | This paper | qPCR primer | GCAACAC GTCCTCA CTACTT |
| Sequence-based reagent | gln-1 R | This paper | qPCR primer | GATTGTT GATTCTG ACGCCAT TT |
| Sequence-based reagent | gdh-1 F | This paper | qPCR primer | AGAGCAG ATGAAGC AAGTCAA G |
| Sequence-based reagent | gdh-1 R | This paper | qPCR primer | CGTCGAT GCCAAGC TCATTAT |
| Sequence-based reagent | con-10 F | This paper | qPCR primer | CTGGCAC TGGTAAC GACAA |
| Sequence-based reagent | con-10 R | This paper | qPCR primer | GCAATTT CGCGCTG TTTCT |
| Sequence-based reagent | flf F | This paper | qPCR primer | GGCAGCG ATAACTC GTGAA |

*Continued*

| Reagent type (species) or resource | Designation | Source or reference | Identifiers | Additional information |
|---|---|---|---|---|
| Sequence-based reagent | *flf R* | This paper | qPCR primer | AAGAAGG CGTAGCA TGTGAA |
| Sequence-based reagent | *pect F* | This paper | qPCR primer | CTTGGGT ATATCAC CGCCTTG |
| Sequence-based reagent | *pect R* | This paper | qPCR primer | CTCCCGA AGGCACA TTGTTA |
| Commercial assay or kit | QuantiTect Reverse Transcription Kit | QIAGEN | Cat#205314 | |
| Commercial assay or kit | Ser/Thr Phosphatase Assay System | Promega | Cat#V2460 | |
| Commercial assay or kit | LightCycler 480 Probes Master | Roche | Cat#048873 01001 | |
| Chemical compound, drug | TriReagent | Sigma-Aldrich | Cat#93289 | |
| Software, algorithm | Statistica 13 | Statsoft Inc, Tulsa, OK, USA | | |
| Software, algorithm | ImageJ | https://doi.org/10.1038/nmeth.2089 | | |
| Other | RNA sequencing data | doi:10.5061/dryad.t4b8gtj4p | | See RNA sequencing and data analysis in Material and Methods |

## Resource availability

### Lead contact
Further information and requests for resources and reagents should be directed to and will be fulfilled by the lead contact, Krisztina Káldi (kaldi.krisztina@med.semmelweis-univ.hu).

### Materials availability
Plasmids and strains generated in this study are available upon request from the lead contact.

## Experimental model and subject details
*Neurospora crassa* strains *wt* (FGSC #2489), *wt,bd* (FGSC #1858), *bd;frq¹⁰* (FGSC #7490), *bd;frq⁹* (FGSC #7779), *rgb-1* (FGSC #8380) and *mcb* (FGSC #7094) were obtained from the Fungal Genetics Stock Center (http://www.fgsc.net/ **McCluskey, 2003**). The last two knockouts were created during the Neurospora Genome Project (**Colot et al., 2006**). To generate the *frq10 Δfcd1-2* strain, *pBM60-ClaI-ΔFCD1-2* (**Querfurth et al., 2011**) was integrated into the *his-3* locus of *bd;frq¹⁰,his-3* by electroporation (**Margolin et al., 1997**). In addition, the *bd; Δwc1* (**Bell-Pedersen et al., 1996**) and the *qa-gsk* (**Tataroğlu et al., 2012**) strains were used in this study.

The standard liquid medium contained Vogel's medium (**Vogel, 1964**) supplemented with 0.5% L-arginine, 10 ng/ml biotin and 2% of glucose. In the starvation medium, the glucose concentration was reduced to 0.01%. For liquid cultures, mycelial mats were grown in Petri dishes in standard liquid medium for 2 days at room temperature in constant darkness. From the mycelial mat discs were punched out, washed by sterile distilled water and transferred into Erlenmeyer flasks containing liquid medium and were grown at 25 °C and shaken with 90 rpm if not indicated otherwise. For growing, at least 150 ml of liquid medium per mycelial discs was used in order to keep glucose concentration as constant as possible during the incubation. Under these conditions *Neurospora* grows as balls of mycelium.

## Method details

### Protein analysis

Extraction of *Neurospora* protein, Western blots (*Gyöngyösi et al., 2013*; *Schafmeier et al., 2006*) and subcellular fractionation (*Gyöngyösi et al., 2017*; *Luo et al., 1998*) was performed as described earlier. In subcellular fractionation same amount of cytosolic and nuclear fractions were analyzed. Ponceau S staining was used as loading control. Representative Western blots are shown. Protein levels were determined by densitometry with the ImageJ software (https://imagej.nih.gov/ij/download.html).

### Race tube assay

Race tubes containing standard race tube medium (*Gyöngyösi and Káldi, 2014*; *Gyöngyösi et al., 2017*) with 2% glucose were inoculated with mycelia originated from starved or normal cultures. Growth front was marked at specific time points and growth rate was calculated accordingly.

### RNA analysis

Total RNA was extracted using the TriReagent (Sigma Aldrich #93289) isolation reagent and transcript levels were quantified as described earlier (*Gyöngyösi et al., 2013*). Values were normalized to the *wt* grown in 2% glucose containing standard medium in each experiment unless indicated otherwise. As levels of previously used housekeeping gene *actin* decreased under starvation conditions, additional genes were selected based on literature data (*Cusick et al., 2014*; *Hurley et al., 2015*; *Llanos et al., 2015*). We found *gna-3* to be the least variable under the applied experimental conditions. $C_t$ values are presented in *Figure 1—source data 4*. For sequences of oligonucleotides and hydrolysis probes see Key resources table.

### RNA sequencing and data analysis

Liquid cultures were grown under standard and glucose starving conditions in 12–12 hr light-dark cycles for 48 hr and samples were harvested at ZT12 (n=4 for each group). RNA was purified using the TriReagent (Sigma Aldrich #93289) isolation reagent. Following DNase treatment RNA quality was controlled by the Nanodrop One$^C$ spectrophotometer, the Qubit 4.0 Fluorometer (Invitrogen) and the Agilent TapeStation 4150 System.

Library preparation and sequencing were performed by BGI Genomics, China, using PE-100 library. Sequencing quality check was performed with FastQC (*Andrews, 2010*). Mapping was performed with STAR (*Dobin et al., 2013*) to the *N.c.* genome from Ensembl (Neurospora_crassa.NC12.48) (*Yates et al., 2020*), and the indexing and duplicate filtering with samtools (*Li et al., 2009*). Read counting was done using HTSeq-count (*Anders et al., 2015*). Differential expression analysis was done with DESeq2 (*Love et al., 2014*) package in R. (*R Development Core Team, 2020*). (Raw data of the RNA sequencing: https://doi.org/10.5061/dryad.t4b8gtj4p).

For primer sequences used during the validation of RNAseq results see Key resources table.

Regarding *Figure 4—figure supplement 6* and *Figure 4—source data 4* it is important to note that KEGG analysis annotates less genes to the different metabolic pathways than GO analysis.

## Phosphatase assay

Measuring PP2A activity was carried out with a Ser/Thr Phosphatase Assay System (Promega #V2460) according to the manufacturer's instructions. Each reaction contained 10 µg protein and was performed for 20 min at room temperature. Absorbance was measured at 600 nm.

## Quantification and statistical analysis

For statistical analysis, the Statistica 13 (Statsoft Inc, Tulsa, OK, USA) software was used. Error bars indicate ± SEM. Results were considered to be statistically significant when p value was<0.05. (*p<0.05; **p<0.01; ***<0.001; n.s.: non-significant) Further statistical details can be found in the figure legends. N represents the number of independent biological samples.

## Acknowledgements

We thank the Fungal Genetics Stock Center and the Neurospora Genome Project for providing *Neurospora* strains. We also thank Rita Krisztina Nagy and Fanni Kóródi for excellent technical assistance and Ágnes Réka Sűdy for critical reading of the manuscript.

This work was supported by grants of the National Research, Development and Innovation Office, NKFIH (K132393 to KK; FK132474 to NG and FK134267 to OK), the Ministry of Innovation and Technology of Hungary (National Research, Development and Innovation Fund, financed under the project of TKP2021-EGA-25) and by the ÚNKP-19–3-IV-SE-3 New National Excellence Program of the Ministry for Innovation and Technology to AS. MB was funded by the Deutsche Forschungsgemeinschaft (DFG, TRR186).

## Additional information

### Funding

| Funder | Grant reference number | Author |
|---|---|---|
| National Research, Development and Innovation Office | K132393 | Krisztina Káldi |
| National Research, Development and Innovation Office | FK132474 | Norbert Gyöngyösi |
| National Research, Development and Innovation Office | FK134267 | Orsolya Kapuy |
| Ministry of Innovation and Technology | TKP2021-EGA-25 | Krisztina Káldi |
| Ministry of Innovation and Technology | ÚNKP-19-3-IV-SE-3 | Anita Szőke |
| Deutsche Forschungsgemeinschaft | TRR186 | Michael Brunner |

The funders had no role in study design, data collection and interpretation, or the decision to submit the work for publication.

### Author contributions

Anita Szőke, Data curation, Formal analysis, Investigation, Visualization, Methodology, Writing – original draft, Project administration, Writing – review and editing; Orsolya Sárkány, Formal analysis, Investigation; Géza Schermann, Data curation; Orsolya Kapuy, Conceptualization; Axel CR Diernfellner, Investigation; Michael Brunner, Conceptualization, Writing – review and editing; Norbert Gyöngyösi, Conceptualization, Data curation, Formal analysis, Funding acquisition, Investigation, Writing – original draft, Writing – review and editing; Krisztina Káldi, Conceptualization, Data curation, Supervision, Funding acquisition, Writing – original draft, Writing – review and editing

### Author ORCIDs

Anita Szőke ![ORCID] http://orcid.org/0000-0003-0593-2088
Krisztina Káldi ![ORCID] http://orcid.org/0000-0002-5724-0182

### Decision letter and Author response

Decision letter https://doi.org/10.7554/eLife.79765.sa1
Author response https://doi.org/10.7554/eLife.79765.sa2

## Additional files

### Supplementary files
- MDAR checklist
- Source data 1. Raw source data images to *Figures 1–3*.
- Source data 2. Source data images with labels of *Figures 1–3*.

### Data availability
All data generated or analysed during this study are included in the manuscript and supporting file; Source Data files have been provided for figures containing Western blot images; Sequencing data have been deposited in Dryad: https://doi.org/10.5061/dryad.t4b8gtj4p.

The following dataset was generated:

| Author(s) | Year | Dataset title | Dataset URL | Database and Identifier |
| --- | --- | --- | --- | --- |
| Szőke A, Sárkány O, Schermann G, Kapuy O, Diernfellner A, Brunner M, Gyongyosi N, Káldi K | 2022 | Adaptation to starvation requires a flexible circadian clockwork in Neurospora crassa (RNAseq) | https://doi.org/10.5061/dryad.t4b8gtj4p | Dryad Digital Repository, 10.5061/dryad.t4b8gtj4p |

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
