## [Editor Report]

This manuscript is of interest to researchers working in the areas of chronobiology, metabolism, or environmental adaptation mechanisms. The authors show that starvation decreases the abundance of the fungal circadian clock component white collar complex (WCC). However, neither phase nor the amplitude of the RNA oscillation of the critical circadian clock gene *frq* are affected by starvation, indicating a mechanism that recalibrates the central clockwork. Furthermore, *Neurospora* recovers faster from starvation in the presence of a functioning clock, adding further evidence for the importance of the circadian clock for organismal fitness.

---

## [Decision Letter]

**Decision letter after peer review:**

Thank you for submitting your article "Adaptation to starvation requires a flexible circadian clockwork in *Neurospora crassa*" for consideration by *eLife*. Your article has been reviewed by 2 peer reviewers, and the evaluation has been overseen by a Reviewing Editor and Jonathan Cooper as the Senior Editor. The reviewers have opted to remain anonymous.

As you can also see from the individual reviews below, our opinion is that the conclusions that could be drawn from the experiments differ substantially from the conclusions currently presented in the manuscript.

In our opinion the key messages are:

a) WC-1,2 are required for the adaptation to starvation in Neurospora and provide a potential link between metabolism, the circadian clock, and light. However, which of the different roles of the WCC is required for this remains unclear. It could be the light receptor, circadian clock, or a light/clock independent transcriptional regulation function.

Here the authors should also reconcile the somewhat contradictory findings that starvation leads to strong decreases in WCC levels, but the transcriptional and growth response to starvation requires WCC.

b) The severe downregulation of WC-1 and WC-2 protein levels and WCC transcriptional activity by starvation, but the simultaneous lack of effects on frq RNA and protein levels is not explainable by the conventional TTL model of the Neurospora circadian clock.

(Here the authors could use *eLife*'s "Ideas and Speculation" subsection to hypothesize how this could work.)

Although we request significant changes in the manuscript's conclusions, we nevertheless find these results (and resulting conclusions) scientifically interesting enough to potentially merit publication in *eLife*.

As essential revisions, we thus ask the authors to carefully re-inspect their results and avoid biased interpretations or overinterpretations. Consider another title that is more representative of the results presented in the manuscript.

Besides the re-interpretation of the data, another essential revision is that the data in Figure 2 must be analyzed quantitatively, as those are critical for the manuscript's conclusions.

Please also carefully consider the major comments made below. As the requested revisions are substantial the revised manuscript will be sent for re-review.

*Reviewer #1 (Recommendations for the authors):*

The text is well-written, with very few typos or grammatical errors.

Page 4 Line 10: The reference to Adhvaryu does not seem appropriate to support the statement it is attached to.

P5 L1-2: Is there a source for the claim that constant light results in steady-state levels of frq RNA and protein?

P7 L 4 and Figure S2: The fluffy data are not convincing. There is only one cycle assayed and the levels simply decline. At least 2 cycles are needed for a convincing rhythm.

P7 L17 and Figure 3A: The finding of no change in WC-1 levels in frq9 on starvation is not convincing. Levels of WC-1 are already very low in frq9 on high glucose and the finding that they do not drop further on starvation may simply indicate that a constitutive background level of WC-1 expression is all that is present in frq9 and it can't drop lower on starvation.

P12 L5: Low growth rate of frq10 on a solid medium is not the usual phenotype – frq10 grows as fast as, or faster than, wild type. I would be very suspicious of the genotype of this strain if it does not grow well on normal solid agar. If it grew well on a normal medium but did not grow after starvation, that is an observation worth reporting.

P16 L2-10: Correct nomenclature for multiple mutant strains should be used, with mutations in order of linkage group and semicolons between linkage groups (see Perkins 1999, 1.9. Multilocus genotypes: https://www.fgsc.net//fgn46/perkins.htm).

P16 L7-9: Is it necessary to thank an author of the paper for providing strains?

P17 L9-10: How did you "find gna-3 to be the least variable"? Did you test this yourself or use the published data?

P32 L19 and 21: Left panel and Right panel should be Upper and Lower panels?

P36 Figure 5C: This experiment is missing an important control. What would the growth of these strains be without any starvation? Without this control, we don't know if the "recovery" growth is the same as growth in high glucose without prior starvation.

P36 Figure 5D: The race tube method is unclear. The agar medium is not specified. How many days of growth are we looking at on the race tubes? The Y axes of the graphs should use the same scale, starting at zero and ending at the same value.

P37 L16: Are the race tubes really 0.2% glucose and not 2%?

P41 L3: Does this journal have STAR methods?

P48 and P52 and P54, Suppl Table 2 and 3 and 4: There doesn't seem to be a supplemental table 1.

*Reviewer #2 (Recommendations for the authors):*

More clarity could be achieved by describing the logic for the different experimental protocols and conditions. I think much of my confusion came from the order of the Figures, might not Figure 2 be better first, describing the effect of starvation on clock function, before Figure 1 which I a fairly detailed investigation of the effect on light induction?

Page 9 "However, while starvation did not alter frq levels in wt, it decreased the amount of frq9 RNA (Figure 3B). This suggests that FRQ contributes to both WCC depletion and FRQ level compensation." I really did not understand this, do you mean that because the RNA did not change it must be something to do with the protein? Is not an alternative explanation that FRQ is not involved in the starvation response and the changes in WCC are due to another system? This needs to be clarified. My interpretation of the data is that FRQ is not involved in the changes in WCC in response to starvation.

Page 9 "As shown in Figure 3A, absence of a 17 functional FRQ protein attenuated the effects of starvation, i.e. amount of WC-1 did not change in response to glucose deprivation, whereas WC-2 levels were moderately reduced." Isn't likely that a reduction in WC1 and WC2 levels could not be measured in response to starvation because they are both already so low in frq9 mutant? Again I feel that the evidence that FRQ participates in the starvation of WC abundance is not compelling.

Please provide a more detailed analysis of the transcriptional responses providing an indication of the regulatory pathway, ie is it direct from WCC to the regulation of the transcripts, or are there intermediatory steps?

I think the Discussion would be improved by a comparison of the effects of glucose on the Neurospora clock with the effects of glucose found in mice and Arabidopsis.

As far as I understand there are three main conclusions, starvation reduces WCC, the clock is able to run despite the uncoupling of WCC from FRQ, and that WCC is required for the full starvation response. However, the Discussion section lacks an explanation of the apparent contradiction of sucrose decreasing WCC, but WCC is needed for the transcriptional response. Please can you address this?

---

## [Author Response]

Reviewer #1 (Recommendations for the authors):The text is well-written, with very few typos or grammatical errors.Page 4 Line 10: The reference to Adhvaryu does not seem appropriate to support the statement it is attached to.

We thank you for the correction and removed the reference.

P5 L1-2: Is there a source for the claim that constant light results in steady-state levels of frq RNA and protein?

We added the following references to the text: Crosthwaite et al., 1995, Cell; Elvin et al., 2005, GD. Crosthwaite et al. showed that frq RNA stops oscillating in constant light. In Elvin et al. both RNA and protein levels were analyzed for 48 hours in LL following a dark-light transfer. frq RNA did not display further changes after 1-4 hours in light, whereas FRQ protein levels gradually increased during the first 8-12 hours and remained high for the rest of the detection period.

P7 L 4 and Figure S2: The fluffy data are not convincing. There is only one cycle assayed and the levels simply decline. At least 2 cycles are needed for a convincing rhythm.

In the revised version of the manuscript fluffy expression is shown for the same period as the expression of frq and ccg-2. We added a reference (Belden et al., 2007, GD) regarding the expected time points of peak and trough of fluffy expression.

P7 L17 and Figure 3A: The finding of no change in WC-1 levels in frq9 on starvation is not convincing. Levels of WC-1 are already very low in frq9 on high glucose and the finding that they do not drop further on starvation may simply indicate that a constitutive background level of WC-1 expression is all that is present in frq9 and it can't drop lower on starvation.

We agree with the reviewer that the low expression of the WCC in frq^9^ (even in the high glucose medium) makes it difficult to compare the glucose dependence of WCC expression in frq^9^ and wt. The conclusion has been modified and the new version of the text focuses mainly on the strain-dependent difference in the changes of frq RNA expression (P7 L22-P8 L14; P16 L21 – P17 L6).

P12 L5: Low growth rate of frq10 on a solid medium is not the usual phenotype – frq10 grows as fast as, or faster than, wild type. I would be very suspicious of the genotype of this strain if it does not grow well on normal solid agar. If it grew well on a normal medium but did not grow after starvation, that is an observation worth reporting.

We regret the inaccurate formulation in the former version of the manuscript. Growth rate of frq^10^ was largely variable on solid medium in our first experiments, especially at the beginning of the race tubes, therefore we disregarded their evaluation. For the revised version we repeated the experiment with a larger sample size (n=12) and focused on shorter (6-hour) periods at the beginning of the tube. Results of this experiment are now shown in (Figure 5 —figure supplement 1).

In addition, we measured the growth rate of frq^10^ inoculated from non-starved cultures and found it similar to that of wt (Author response image 1).

**Author response image 1. sa2fig1:** Growth rate on race tubes containing 2% glucose. Cultures were incubated in 2% liquid medium before the inoculation to race tubes. (n=15; n.s.).

P16 L2-10: Correct nomenclature for multiple mutant strains should be used, with mutations in order of linkage group and semicolons between linkage groups (see Perkins 1999, 1.9. Multilocus genotypes: https://www.fgsc.net//fgn46/perkins.htm).

We thank you for the comment and corrected the nomenclature in the Materials and methods.

P16 L7-9: Is it necessary to thank an author of the paper for providing strains?

We appreciate the comment and corrected the section in the Materials and methods accordingly.

P17 L9-10: How did you "find gna-3 to be the least variable"? Did you test this yourself or use the published data?

As starvation resulted in a decrease of actin expression even in wt, i.e. cycle times in real-time PCR changed characteristically, we looked for other possible housekeeping genes in the literature which were described as non-oscillating and least variable under different metabolic conditions (Cusick et al., 2014; Hurley et al., 2015; Llanos et al., 2015). Tfc1, SarA and gna-3 were tested as possible reference genes. gna-3 was selected, as its levels remained practically constant despite changes in the glucose content of the medium (Figure 1 – Source data 14 (please find in the Article File: P42 L3-7)).

We completed this section in the Materials and methods with this information: "As levels of previously used housekeeping gene actin decreased under starvation conditions, additional genes were selected based on literature data (Cusick et al., 2014; Hurley et al., 2015; Llanos et al., 2015). We found gna-3 to be the least variable under the applied experimental conditions. C_t_ values are presented in Figure 1 – Source data 14. For sequences of oligonucleotides and hydrolysis probes see Key resources table.

P32 L19 and 21: Left panel and Right panel should be Upper and Lower panels?

We thank you for the correction and changed the text accordingly.

P36 Figure 5C: This experiment is missing an important control. What would the growth of these strains be without any starvation? Without this control, we don't know if the "recovery" growth is the same as growth in high glucose without prior starvation.

We agree with the criticism and in the revised version of the manuscript we show the result of this control experiment: Non-starved control samples did not show strain-specific differences in the acceleration of growth after the transfer into fresh glucose-rich medium (Figure 5C).

P36 Figure 5D: The race tube method is unclear. The agar medium is not specified. How many days of growth are we looking at on the race tubes? The Y axes of the graphs should use the same scale, starting at zero and ending at the same value.

For the revised version of the manuscript we repeated the race tube measurements in a modified form allowing the analysis of growth rate for shorter (6 and 12-hour-) periods. The sample size was enlarged and frq^10^ was also included. The new diagram shows growth rates normalized to those of the non-starved samples (inoculated from non-starved cultures onto the race tubes and indicated with dashed line) of the same strain during the same period. In addition, a more detailed description of the experiment was added to "Figure 5 —figure supplement 1" and a subsection was added to Materials and methods.

P37 L16: Are the race tubes really 0.2% glucose and not 2%?

The race tube medium in the experiments shown in the previous version of the manuscript contained 0.2 % glucose. This is the usual glucose concentration when the conidiation rhythm is analyzed. However, to achieve a more characteristic increase in the glucose level, in the experiments shown in the revised version we used race tubes with 2 % glucose.

P41 L3: Does this journal have STAR methods?

Thank you for the feedback, we amended the wording to ‘Materials and methods section, which is the correct title for this section in *eLife*.

P48 and P52 and P54, Suppl Table 2 and 3 and 4: There doesn't seem to be a supplemental table 1.

Because of the size limitation of the journal, Figure 4 – Source data 2 (in the previous version of the manuscript entitled as Supplementary Table 1) was uploaded as an excel file and shows data from the Gene Ontology enrichment analysis.

Reviewer #2 (Recommendations for the authors):More clarity could be achieved by describing the logic for the different experimental protocols and conditions. I think much of my confusion came from the order of the Figures, might not Figure 2 be better first, describing the effect of starvation on clock function, before Figure 1 which I a fairly detailed investigation of the effect on light induction?

To better justify the choice of different protocols (e.g. LL, LD and DD conditions), we integrated more explanation into the description of the experimental procedures in the Results section (e.g. P5 L7-L10). Based on the suggestion of the reviewer we also changed the order of Figure panels. In the revised version Figure 1 shows expression of clock proteins and RNAs in both LL and DD, while Figure 2 summarizes the data concerning the light induction of frq, wc-1 and al-2 transcription and the subcellular localization of the clock components.

Page 9 "However, while starvation did not alter frq levels in wt, it decreased the amount of frq9 RNA (Figure 3B). This suggests that FRQ contributes to both WCC depletion and FRQ level compensation." I really did not understand this, do you mean that because the RNA did not change it must be something to do with the protein? Is not an alternative explanation that FRQ is not involved in the starvation response and the changes in WCC are due to another system? This needs to be clarified. My interpretation of the data is that FRQ is not involved in the changes in WCC in response to starvation.

After rereading the criticized section, we admit that the text was not well structured and we carried out several modifications. We intended to emphasize that upon drastic changes of the glucose availability frq RNA levels remained compensated in wt, but this compensation was affected when functional FRQ was not present. We agree with the reviewer's opinion that the low expression of the WCC in frq^9^ makes it difficult to compare the glucose-dependence of WCC expression in frq^9^ and wt. We modified the conclusion by adding this information and now mainly focus on the strain-dependent difference in the changes of frq RNA expression. (P7 L22-P8 L14)

Page 9 "As shown in Figure 3A, absence of a functional FRQ protein attenuated the effects of starvation, i.e. amount of WC-1 did not change in response to glucose deprivation, whereas WC-2 levels were moderately reduced." Isn't likely that a reduction in WC1 and WC2 levels could not be measured in response to starvation because they are both already so low in frq9 mutant? Again I feel that the evidence that FRQ participates in the starvation of WC abundance is not compelling.

As mentioned in the previous answer, we agree with the concerns of the reviewer and in the criticized section we modified the conclusion (P8 L11-L14). We emphasize in the new version of this section that changes in frq RNA expression (frq promoter activity) are mainly different in wt and frq^9^.

Please provide a more detailed analysis of the transcriptional responses providing an indication of the regulatory pathway, ie is it direct from WCC to the regulation of the transcripts, or are there intermediatory steps?

We now show the list of genes (Figure 4 – Source data 1) that changed in a strain-specific manner in response to glucose starvation and, based on ChIP-seq results, were earlier described as direct targets of the WCC (Smith et al., 2010; Hurley et al., 2014). Based on the literature data showing that the WCC affects the expression of several other transcription factors and controls basic cellular functions which might affect the expression of further genes, it was not surprising that only 90 out of the 1377 genes were reported to be direct targets of the WCC.

I think the Discussion would be improved by a comparison of the effects of glucose on the Neurospora clock with the effects of glucose found in mice and Arabidopsis.

We included a short section into the Discussion which gives a short overview about known interactions between glucose availability and circadian timekeeping in different organisms (P15 L18-P16 L7).

As far as I understand there are three main conclusions, starvation reduces WCC, the clock is able to run despite the uncoupling of WCC from FRQ, and that WCC is required for the full starvation response. However, the Discussion section lacks an explanation of the apparent contradiction of sucrose decreasing WCC, but WCC is needed for the transcriptional response. Please can you address this?

We agree with the reviewer that the problem of how low levels of WCC could sufficiently support the transcription of frq and different output genes under starvation conditions was not discussed properly. Our results suggest a model in which the maintained level of nuclear WCC and the weakened inhibition by both FRQ (the hyperphosphorylated form is less active in the negative feedback) and PKA (its activity lowered upon glucose depletion) together might ensure that transcriptional activity of the WCC is preserved upon glucose withdrawal in both DD and LL despite the decrease of the overall level of the complex. In the revised version these aspects are discussed more thoroughly (P16-18).